# AI-Generated Video Detection via Perceptual Straightening

**Christian Internò**[1,3*]   **Robert Geirhos**[2]   **Markus Olhofer**[3]   **Sunny Liu**[4]
**Barbara Hammer**[1]   **David Klindt**[4]

[1] Bielefeld University   [2] Google DeepMind
[3]Honda Research Institute EU   [4]Cold Spring Harbor Laboratory

## Abstract

The rapid advancement of generative AI enables highly realistic synthetic videos, posing significant challenges for content authentication and raising urgent concerns about misuse. Existing detection methods often struggle with generalization and capturing subtle temporal inconsistencies. We propose ***ReStraV**(Representation Straightening for Video)*, a novel approach to distinguish natural from AI-generated videos. Inspired by the *"perceptual straightening"* hypothesis [1, 2]—which suggests real-world video trajectories become more straight in neural representation domain—we analyze deviations from this expected geometric property. Using a pre-trained self-supervised vision transformer (DINOv2), we quantify the temporal curvature and stepwise distance in the model's representation domain. We aggregate statistics of these measures for each video and train a classifier. Our analysis shows that AI-generated videos exhibit significantly different curvature and distance patterns compared to real videos. A lightweight classifier achieves state-of-the-art detection performance (e.g., 97.17% accuracy and 98.63% AUROC on the VidProM benchmark [3]), substantially outperforming existing image- and video-based methods. *ReStraV* is computationally efficient, offering a low-cost and effective detection solution. This work provides new insights into using neural representation geometry for AI-generated video detection.

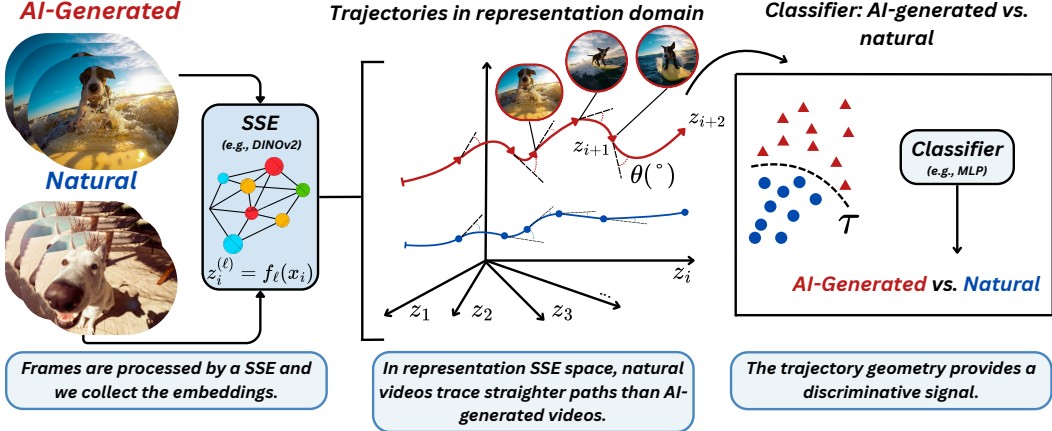

Figure 1: **The ReStraV method for AI-video detection.** Inspired by "perceptual straightening," our approach leverages the geometric insight that natural videos form "straighter" feature trajectories ($z_i$) than generated ones. The temporal curvature (Eq. 1) serves as the discriminative signal for detection.

*Correspondence to: `christian.interno@uni-bielefeld.de`

**Code:** `https://github.com/ChristianInterno/ReStraV`

39th Conference on Neural Information Processing Systems (NeurIPS 2025).

# 1 Introduction

Generative AI has significantly advanced in synthesizing realistic video content [4–6]. Early approaches (e.g., generative adversarial networks, variational autoencoders) struggled with fidelity and temporal coherence [7–9]. However, rapidly evolving large-scale foundation models have introduced sophisticated generative techniques [10, 11]. These methods, often diffusion models [12] and transformer-based architectures [13, 14], can produce near-photorealistic videos from text or initial frames. As these systems improve, the ability to easily generate convincing synthetic videos raises pressing concerns about malicious manipulation and fabricated visual media [15].

Robust strategies to detect AI-generated content are therefore urgently needed [6, 11, 16]. Detecting AI videos is more challenging than AI generated image due to temporal consistency requirements that necessitate thorough analysis across frames [3, 17]. Traditional deepfake detectors, often tuned to specific artifacts (e.g., face-swapping irregularities), may not generalize to diverse generative methods [18]. Moreover, large-scale pretrained foundation encoders may not explicitly learn features optimized for AI content detection. Watermarking is one option but relies on model operators' goodwill and can be circumvented [19–22]. Thus, detection methods are needed that capture AI generation anomalies, regardless of the underlying generative approach.

This work explores neural representational distance and curvature (formally defined in Eq. (1)) as discriminative signals for fake video detection. An overview of *ReStraV* is provided in Fig. 1. According to the *perceptual straightening hypothesis*, natural inputs map to straight paths in neural representations while unnatural sequences form curved trajectories [1, 2]. This has been verified in neuroscience, psychophysics, on CNNs and LLMs [1, 23]. It is motivated by the idea that predictive coding might favor straight temporal trajectories in latent space because they are more predictable.

Taking inspiration, in this work, we hypothesize a distinction between natural and AI generated videos in artificial neural networks (ANNs). While ANNs may not perfectly replicate biological straightening [1, 2, 24], we expect their learned representations to show AI-generated videos as more curved in activation space than real videos. We surmise synthetic videos exhibit curvature patterns deviating from the lower curvature trajectories of real events, supported by differing ANN representational dynamics for natural versus artificial videos [24], as illustrated in Fig. 2A and Fig. 4.

To test this hypothesis, we use the DINOv2 ViT-S/14 pretrained visual encoder [25], chosen for its sensitivity to generative artifacts (Fig. 2B). For each video, we extract frame-level complete set of patch embeddings and Classify token (CLS) from DINOv2's final transformer block (`block.11`). From this trajectory, we quantify local curvature (angle between successive displacement vectors, measuring path bending) and stepwise distance (change magnitude between consecutive frame representations), as in Eq. (1). We then derive descriptive statistics (mean, variance, min, max; examples in Fig. 5) from these per-video time series of curvature and distance. These aggregated geometric features (Section 5) are used by a lightweight classifier (Section 6) to distinguish real from AI content.

*ReStraV* re-purposes DINOv2 as a "feature space" for temporal anomalies. DINOv2's extensive training on natural data provides a latent space where real video trajectories should be characteristically smooth or "straight" (Fig. 2A, Fig. 4). Deviations, like increased jitter or erratic curvature often in AI videos, become discernible geometric signals of synthetic origin. Importantly, *ReStraV* is computationally efficient, processing videos in approximately 48 ms end-to-end (including DINOv2 forward pass). *ReStraV* is thus a low-cost alternative to resource-intensive methods (details in Section 6). By exploiting ANN's activation dynamics, *ReStraV* offers a simple, interpretable AI-video detection approach (experimental validation in Section 7). Our contributions are as follows:

1. We propose a novel, simple, cost-efficient, and fast representational geometry strategy for AI-generated video detection, leveraging neural activation distance and curvature as reliable indicators of generated videos.

2. We show the approach yields a reliable "fake video" signal across vision encoders, even those not trained on video data; DINOv2's [25] self-supervised representations excel without task-specific tuning.

3. We demonstrate through extensive experiments on diverse benchmarks (VidProM [3], GenVidBench [17], and Physics-IQ [26]) and models, that *ReStraV* improve detection accuracy that often surpasses state-of-the-art (SoTA) methods.

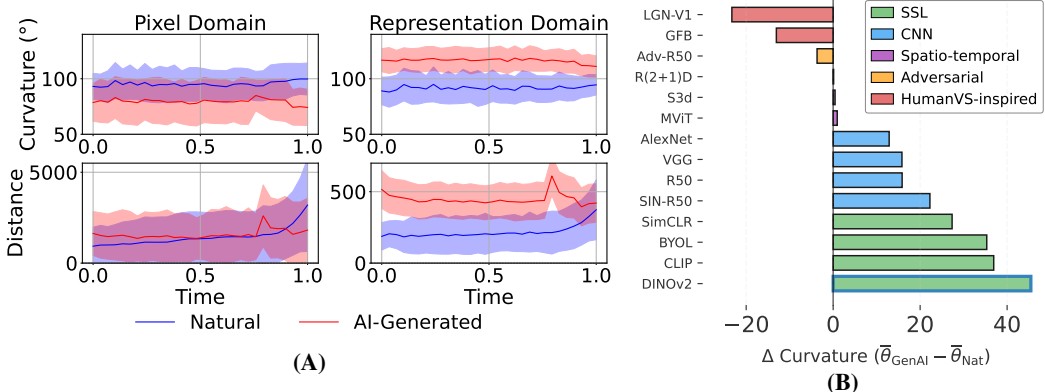

Figure 2: **(A)** In pixel space (left), video trajectory metrics (curvature, distance; see Eq. (1) for details) between natural vs. AI-generated videos show substantial overlap. In contrast, DINOv2 representations (right) straighten natural trajectories, clearly separating natural and AI-generated videos. **(B)** The mean curvature gap ($\Delta\theta$) between AI-generated and natural videos across various visual encoders. HVS-inspired models (red) exhibit negative deltas, straightening both natural and AI videos equally, while SSL models (green), particularly DINOv2, show the largest positive deltas.

## 2 Related work: detecting AI–generated videos

Detecting AI-generated video (see Appendix A for an AI generative models overview) is becoming increasingly challenging. Many early detection efforts, including image-based detectors (CNNSpot [27], Fusing [28], Gram-Net [16], FreDect [29], GIA [30], LNP [31], DFD [32], UnivFD [33]), focused on spatial or frequency-domain artifacts within individual frames. However, their frame-centric nature limits their efficacy on videos, where temporal consistency is paramount.

Dedicated video detectors, such as adapted action recognition models (TSM [34], I3D [35], Slow-Fast [36]) and Transformer-based (X3D [37], MVIT-V2 [38], VideoSwin [39], TPN [40], UniFormer-V2 [41], TimeSformer [42], DeMamba [43], aim to learn motion anomalies and temporal inconsistencies. While advancing temporal modeling, they require extensive training and may still struggle across rapidly evolving AI generative models. Those approaches may overlook a more fundamental signal: geometric distortions in the temporal trajectory of neural representations. We hypothesize that the geometric properties of these trajectories—reflecting the inherent smoothness and predictability of natural dynamics that generative models fail to replicate—offer a more robust signal for detection. Unlike related work in video quality assessment that also uses trajectories [44, 45], our focus is distinctly on detecting synthetic content, regardless of its perceptual quality.

## 3 Perceptual straightening definition

Natural input sequences are often highly complex. For instance, even a video of a simple object moving across an image will be a nontrivial sequence of points traveling through a high dimensional pixel space. Specifically, this sequence will be curved since the only straight video is an interpolation between two frames. According to the temporal straightening hypothesis, biological visual systems simplify the processing of dynamic stimuli by transforming curved temporal trajectories into straightened trajectories of internal representations [1, 2]. Although the raw pixel trajectories of natural videos are highly curved, the neural representations in the human visual system become straightened to support efficient temporal prediction and processing[2]. In this article, we exploit this property to detect differences between AI-generated and natural videos.

Formally, let a video segment be represented by a temporal sequence of $T$ feature vectors, $\mathcal{Z} = (z_1, z_2, \dots, z_T)$, where each $z_i \in \mathbb{R}^D$ is the embedding for the $i$-th sampled frame (with $i$ being the frame index). The displacement vector between consecutive frame representations is defined as $\Delta z_i = z_{i+1} - z_i$, for $i = 1, \dots, T-1$. The magnitude of this displacement, which we term the stepwise distance, is $d_i = \|\Delta z_i\|_2$. Following [1, 2], the curvature $\theta_i$ of the representation trajectory is defined as the angle between successive displacement vectors, $\Delta z_i$ and $\Delta z_{i+1}$:

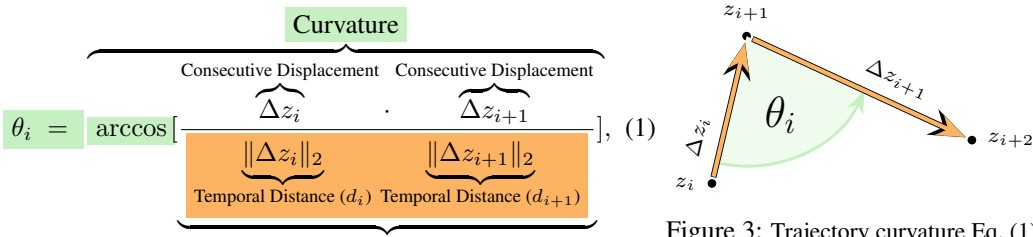

$$\theta_i = \arccos\left[\underbrace{\underbrace{\Delta z_i}_{\|\Delta z_i\|_2} \cdot \underbrace{\Delta z_{i+1}}_{\|\Delta z_{i+1}\|_2}}\right], \quad (1)$$

Figure 3: Trajectory curvature Eq. (1).

$i = 1, \ldots, T - 2$. The curvature $\theta_i$ (Equation 1), defined for each discrete step $i$ along the trajectory (ranging from 1 to $T-2$, where $T$ is the total number of sampled frames), is computed from the cosine similarity between successive displacement vectors $\Delta z_i$ and $\Delta z_{i+1}$. This geometric relationship is visualized in Figure 3. The figure depicts three consecutive frame embeddings ($z_i, z_{i+1}, z_{i+2}$) from the overall dashed trajectory. The orange vectors $\Delta z_i$ and $\Delta z_{i+1}$ represent the displacements between these embeddings, with their respective lengths being the stepwise distances $d_i$ and $d_{i+1}$. The green angle $\theta_i$ shows the turn at $z_{i+1}$. This angle, typically converted to degrees ($\theta_i^\circ = \theta_i \times \frac{180}{\pi}$), provides a measure of how sharply the representation trajectory bends at each step. These metrics, stepwise distance $d_i$ and curvature $\theta_i$, form the core of our geometric analysis.

## 4 Perceptual straightening of natural videos in DINOv2

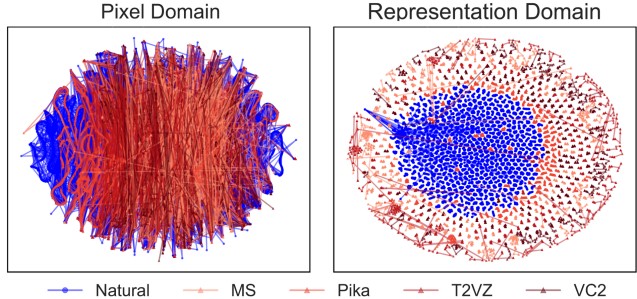

Figure 4: t-SNE embeddings of curvature trajectories for 1,000 videos from the VideoProM dataset [3]: 500 natural and 500 AI-generated (125 each from Pika [46], VideoCrafter2 [47], Text2Video-Zero [48], and ModelScope [49]; 24 frames/video). **Left (Pixel Space):** Natural and synthetic trajectories overlap significantly. **Right (DINOv2 ViT-S/14 Representation Space):** Trajectories clearly separate, with natural (blue) and AI-generated (shades of red) videos forming distinct clusters.

The classic finding is that visual system model have lower curvature in the representation space compared to pixel space–this is called *perceptual straightening* [1, 2]. For our application, we want to compare the relative curvature of natural and AI generated videos in any representational space. Just for simplicity, we could call this *real straightening* as in real videos are less curved than AI videos. One might expect that a model with good *perceptual straightening* also has good *real straightening*.

To test this hypothesis, we analyze fourteen vision encoders across diverse families: Supervised CNNs (AlexNet [50], VGG-16 [51], ResNet-50 [52], and the texture-debiased SIN-ResNet-50 [53]); Self-Supervised (SimCLR-R50 [54], BYOL-R50 [55], CLIP [56], and DINOv2 [25]); Human Visual System-Inspired (a Gabor Filter Bank [57] and the LGN-V1 model [1]); Spatio-temporal (S3d [58], R(2+1)D [59], and MViT [60]); and an Adversarially Trained ResNet-50 [61]. Surprisingly, as shown in Fig. 2B, we observe that the opposite seems to be the case: good perceptual straighteners actually make natural videos more curved than AI videos. HVS-inspired models achieve the strongest absolute straightening but do so indiscriminately for both real and AI videos, resulting in a negative curvature gap ($\Delta\theta < 0$). In contrast, self-supervised models like DINOv2 reduce the curvature of natural videos, which align with their learned priors of real-world statistics, but do not regularize the trajectories of AI-generated videos, which violate these priors. This differential response creates a large, positive curvature gap ($\Delta\theta = 45.46°$ for DINOv2), which is the foundation of our method's success. The negligible correlation between absolute straightening and detection capability ($\rho = -0.13$, $p = 0.64$) confirms that detection performance hinges not on absolute straightening capability,

but on *differentially* straightening natural versus synthetic videos. While artificial neural networks may not fully replicate the absolute perceptual straightening observed in the biological visual system [24], this *relative* effect is the key mechanism for our detection method.

This principle is clearly visualized in Fig. 2A. Using matched pairs of real videos from Physics-IQ [26] and their AI-generated replicas, we see that geometric metrics overlap considerably in raw pixel space. However, in DINOv2's representation space, the trajectories separate distinctly, offering a clean signal for detection. Further evidence is provided in Fig. 4, which shows that this separation holds on a larger diverse dataset (VideoProM [3]). The t-SNE embeddings of curvature trajectories show natural videos (blue) forming a tight cluster, well-separated from the clusters of AI-generated videos (shades of red). This demonstrates that DINOv2's features effectively surface temporal inconsistencies without any task-specific training. Refer to Appendix A.1 for trajectories samples.

To implement this, we extract $T = 24$ frames $\mathcal{Z} = (z_1, \ldots, z_{24})$ by sampling over a 2-second video duration. This 2-second window is suitable for the videos considered in Section 7 ($\approx 2 - 5$s long with $12 - 30$ FPS), with an temporal based sample frame of $\Delta t = 2s/(24 - 1)$. Nevertheless, we hypothesize that using longer videos could further enhance performance. Our choice of a $2s$ window with 24 frames was found to provide an optimal trade-off between high detection accuracy and computational efficiency, as validated in our ablation studies (see B for details). Each $x_i$ is resized to $224 \times 224$ pixels and normalized to $[0, 1]$. These preprocessed frames are then encoded by the DINOv2 ViT-S/14 model [25]. The 384 CLS (Classify) tokens and 196 patch embedding ($16 * 16$ patches of the $224 * 224$ inputs) from its final transformer block (`block.11`). These token embeddings are then flattened and concatenated to form a single feature vector $z_i \in \mathbb{R}^{75648}$. The sequence of these vectors, $\mathcal{Z}$, forms the temporal trajectory in DINOv2's representation space, from which temporal curvature and distance metrics are computed (Eq. (1)).

> **Takeaway 1:** By projecting videos into DINOv2's representation space, geometric trajectory features (curvature & distance, Eq. 1) become indicators of synthetic origin, differentiating AI-generated videos from natural ones in a way that is not possible in raw pixel space.

## 5 Analyzing characteristics of perceptual trajectories

In order to analyze the differences of natural video trajectory signals vs. AI-generated ones we defined statistical features as the first four descriptive moments of both distance and curvature: mean, minimum, maximum and variance. This yields an 8-dimensional feature vector: $\left[ \mu_d, \min d, \max d, \sigma_d^2, \mu_\theta, \min \theta, \max \theta, \sigma_\theta^2 \right]$, where $\mu_d = \frac{1}{T-1} \sum_i d_i$ and $\sigma_d^2 = \frac{1}{T-1} \sum_i (d_i - \mu_d)^2$ (analogously for curvature $\theta_i^\circ$).

We select 50,000 AI-generated samples (10,000 each from Pika [46], VideoCraft2 [62], Text2Video-Zero [63], ModelScope [3], and Sora [64]) from VideoProM [3]. Concurrently, 50,000 natural videos are randomly chosen from DVSC2023 [47]. All videos are DINOv2 (ViT-S/14) encoded, and their aggregated statistical features are computed. Fig. 5 illustrates these aggregated feature distributions. The top row shows distance features ($\mu_d, \min_d, \max_d, \sigma_d^2$), characterizing inter-frame change magnitude and variability. The bottom row presents corresponding curvature features ($\mu_\theta, \min_\theta, \max_\theta, \sigma_\theta^2$), reflecting angular changes between consecutive frame transitions.

Statistical tests further confirm these observed differences. A two-sample t-test comparing the mean per-video $\mu$ between natural and AI-generated videos produced highly significant results. Distance ($d$): $t = -14.27$, $p = 5.53 \times 10^{-46}$; Curvature ($\theta$): $t = -44.02$, $p \approx 0$. An ANOVA comparing feature distributions among different AI generators and natural videos also shows strong statistical differentiation in DINOv2 embedding space ($F$-value of 18598.17, $p \approx 0$). These observations support our hypothesis: natural videos exhibit smoother, more consistent trajectories (lower $\mu_\theta$, surprisingly higher $\sigma_\theta^2$), while AI-generated videos show irregular transitions resulting in higher curvature metrics but with lower $\sigma_\theta^2$. These differences form the basis for our classification pipeline, where a classifier learns to separate natural from AI videos based on their trajectory geometry.

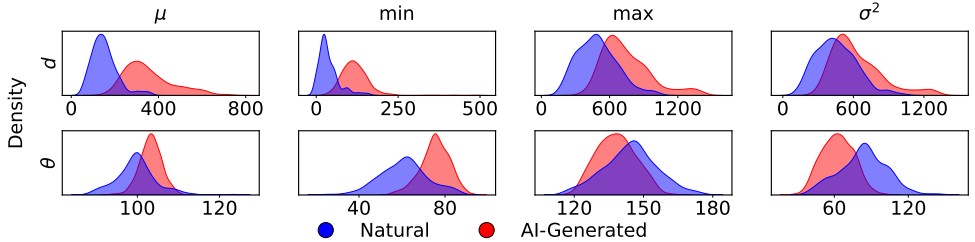

Figure 5: Distributions of aggregated temporal trajectory features (mean, min, max, variance) for natural and AI-generated videos, computed using DINOv2 ViT-S/14 representations. **Top row:** Temporal distance-based features ($d_i$). **Bottom row:** Corresponding curvature-based features ($\theta_i^\circ$). Both distance- and curvature-based features provide discriminative signal.

## 6    Video classifier to detect AI-generated content

Given that DINOv2 representation distance $d$ and curvature $\theta$ differs significantly between natural and AI-generated videos, we evaluate if these features can be used in a lightweight, transparent, and easily replicated classifier without raw pixel processing or DINOv2 fine-tuning. We use the dataset from Section 5 and apply a stratified 50/50 train/test split. Class priors are identical, and each subset is balanced among five AI models (Pika [46], VideoCraft2 [62], Text2Video-Zero [63], ModelScope [3] and Sora [64]). We sample frames and we obtain the signals of distance $\{d_i\}_{i=1}^{T-1}$ and curvature $\{\theta_i^\circ\}_{i=1}^{T-2}$ as detailed in Section 5. For classification, we construct a feature vector $y$ per video by combining direct signals and aggregated statistics from these trajectories. Specifically, $y$ concatenates seven distance values $[d_1, d_2, \ldots, d_7]$ and six curvature values: $[\theta_1^\circ, \theta_2^\circ, \ldots, \theta_6^\circ]$; and four statistical descriptors (mean, variance, minimum, maximum) for both $\{d_i\}$ and $\{\theta_i^\circ\}$. This results in a final feature vector $y \in \mathbb{R}^{21}$. To ensure our curvature-based features are detecting generative artifacts rather than hard scene cut frequency, we performed a robustness analysis detailed in Appendix C.

We consider only off-the-shelf models: logistic regression (LR), Gaussian Naive Bayes (GNB), random forest (RF; 400 trees, depth $\leq 6$), gradient boosting (GB; 200 rounds, learning rate 0.1), RBF-kernel SVM (calibrated by Platt scaling), and a two-layer MLP ($64 \rightarrow 32$). We perform no feature engineering or hyperparameter search beyond a 3-fold grid/random sweep. For each classifier, we optimize the decision threshold $\tau^*$ on the training set to maximize the $F_1$-score. The chosen threshold $\tau^*$ was then applied unchanged to the test set. Inference cost is reported end-to-end (latency $= T_{\text{DINOv2}} + T_{\text{clf}}$), averaged over the test fold on a single NVIDIA RTX-2080 (see Appendix D). A DINOv2 forward pass (ViT-S/14, block 11, 8-frame batch) takes 43.6 ms. This constant is added to each classifier time ($T_{\text{clf}}$) in Table 1.

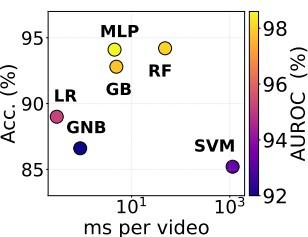

Figure 6: Inference-time ($ms$) vs. accuracy (%) and AUROC (%) for *ReStraV*'s classifiers.

Table 1: Performance and inference time of *ReStraV*'s classifiers, balanced 50k/50k natural/AI-generated video test set from VideoProM [3], cf. Section 5 for details. The best scores are **bold**, second best are underlined and the best method overall is highlighted in **blue**.

| Model | Acc. | Bal. | Spec. | Pr$_{\text{gen}}$ | Re$_{\text{gen}}$ | F1$_{\text{gen}}$ | AUROC | Time (ms) |
|---|---|---|---|---|---|---|---|---|
| SVM | 85.23 | 85.78 | 86.42 | 96.93 | 85.04 | 90.62 | 93.27 | 1183.94 |
| GNB | 86.64 | 84.43 | 81.12 | 95.94 | 87.72 | 91.68 | 92.05 | 44.53 |
| LR | 89.02 | 88.86 | 88.53 | 97.54 | 89.17 | 93.12 | 95.26 | **43.97** |
| GB | 92.83 | 92.31 | 91.57 | 98.25 | 93.16 | 95.63 | 97.85 | 48.59 |
| RF | **94.24** | 88.67 | 80.37 | 96.13 | **97.05** | **96.53** | 98.03 | 48.14 |
| **MLP** | 94.17 | **94.19** | **94.11** | **98.88** | 94.14 | 96.48 | **98.63** | 48.12 |

Table 1 summarizes test performance. The MLP achieves the highest accuracy (94.17%), $F_1$-score (96.48%), and AUROC (98.6%) (Fig. 9A), followed by RF. Section 6 visualizes each classifier's speed/accuracy. Models in the upper-left offer the best cost/benefit. The MLP (highlighted blue in Table 1) achieves top AUROC and $F_1$ while being within $\approx$2ms from GNB. The confusion matrices and ROC (Fig. 9A and B in Appendix A.2) confirms low false positive/negative rates (cf.

Appendix A.2 for decision boundaries visualization). Refer to Appendix A.3 for permutation feature importance analysis.

> **Takeaway 2:** Simple, lightweight classifiers trained directly on geometric trajectory features achieve high classification accuracy ($\approx 94\%$) and AUROC ($\approx 99\%$), offering an efficient detection approach ($\approx 48$ ms per video) without complex modeling.

## 7 Benchmark results

We evaluate *ReStraV* (with MLP from Section 6) under four settings: **A)** Vs. SoTA image-based detectors on VidProM [3]; **B)** Vs. SoTA video detectors on VidProM [3] (evaluating "seen", "unseen", and "future" generator scenarios) and on the GenVidBench [17] dataset (in "cross-source"/"generator" (M) scenarios and its "Plants" hard task subclass (P)); **C)** Extreme generalization tests including one-to-many detection on the DeMamba [43] benchmark [43] and **D)** zero-shot evaluation on the Veo3 model [65]; **E)** Vs. a Vision-Language Model (VLM) performance (Gemini 1.5 Pro [66]) on the Physics-IQ dataset [26] using matched real/generated video pairs. The best scores are **bold**, second best are underlined and the best method overall is highlighted in **blue**.

**A) ReStraV vs. image-based detectors.** We evaluate our method *ReStraV* against eight SoTA image based detectors in Table 2. *ReStraV* is trained with data and processing from Section 6. We use a balanced test set (40,000 real videos; 10,000 AI-generated for each of four models: Pika [46], VideoCraft2 [67], Text2Video-Zero [48], ModelScope [49], replicating [3]'s implementation.

Performance is measured by overall classification accuracy and mean Average Precision (mAP). Table 2 summarizes the results from [3]. Baseline methods achieve moderate accuracies (45%–62%), with LNP [31] and Fusing [28] showing lower values. In contrast, our method obtains 97.06% average accuracy (Pika: 90.90%, VideoCraft2: 99.50%, Text2Video-Zero: 99.05%, ModelScope: 98.37%). *Caveat:* Comparing *ReStraV* to image-based detectors on a video task is not an even comparison (see next section for stronger baselines), yet it highlights the inadequacy of methods that rely solely on image-based features for AI-generated video detection neglecting temporal information.

Table 2: Comparison of *ReStraV* vs. image-based detectors on VidProM [3]. Accuracy (%) (left) and mAP (%) (right). Higher values (darker blue) indicate better performance. ↑ Higher is better.

| Method | Accuracy ↑ (%) | | | | | mAP ↑ (%) | | | | |
|---|---|---|---|---|---|---|---|---|---|---|
| | Pika | VC2 | T2VZ | MS | Avg | Pika | VC2 | T2VZ | MS | Avg |
| CNNSpot [27] | 51.17 | 50.18 | 49.97 | 50.31 | 50.41 | 54.63 | 41.12 | 44.56 | 46.95 | 46.82 |
| FreDect [29] | 50.07 | 54.03 | 69.88 | 69.94 | 60.98 | 47.82 | 56.67 | 75.31 | 64.15 | 60.99 |
| Fusing [28] | 50.60 | 50.07 | 49.81 | 51.28 | 50.44 | 57.64 | 41.64 | 40.51 | 56.09 | 48.97 |
| Gram-Net [16] | 84.19 | 67.42 | 52.48 | 50.46 | 63.64 | 94.32 | 80.72 | 57.73 | 43.54 | 69.08 |
| GIA [30] | 53.73 | 51.75 | 41.05 | 60.22 | 51.69 | 54.49 | 53.21 | 36.69 | 66.53 | 52.73 |
| LNP [31] | 43.48 | 45.10 | 47.50 | 45.21 | 45.32 | 44.28 | 44.08 | 46.81 | 39.62 | 43.70 |
| DFD [32] | 50.53 | 49.95 | 48.96 | 48.32 | 49.44 | 49.21 | 50.44 | 44.52 | 48.64 | 48.20 |
| UnivFD [33] | 49.41 | 48.65 | 49.58 | 57.43 | 51.27 | 48.63 | 42.36 | 48.46 | 70.75 | 52.55 |
| **ReStraV** | **90.90** | **99.50** | **99.05** | **98.37** | **97.06** | **99.12** | **98.76** | **98.93** | **98.44** | **98.81** |

**B) ReStraV vs. video-based detectors.** We firstly compare *ReStraV* against the widely recognized VideoSwinTiny [39] (implementation from [17]) on the VidProM [3], with data setup following Section 7. Our evaluation considers three scenarios: **Seen generators:** Models are trained and tested on videos from a pool of the same four AI generators in Section 7, using a balanced set of 80,000 videos with a 50/50 train/test split. **Unseen generators:** Generalization is assessed by training models while excluding two specific AI generators (e.g., VC2 [62] and T2VZ [48]), which are then used for testing. **Future generators:** To simulate encountering a novel advanced model, *ReStraV* (trained on older generators) is tested on Sora [68]. Accuracy and mAP results in Table 3.

We futher evaluate *ReStraV*'s vs. nine SoTA video based detectors in Table 4. We consider two settings: **Main (M)** task, which is designed to test generalization across generators. Detectors are

trained on videos from Pika [46], VideoCraft2 [67], Text2Video-Zero [48], ModelScope [49], and tested on MuseV [69], Stable Video Diffusion (SVD) [70], CogVideo [71], and Mora [72]. **Plants (P)** task, the most challenging subset from [17]. The challenge may arise from the complex and often stochastic nature (e.g., irregular leaf patterns, subtle wind movements), which can make generative artifacts less distinguishable from natural variations or harder for models to consistently detect (qualitative samples in Appendix A.4). We use the same setting of task (M) but focusing on videos of plants in the test set. Baseline's results from [17].

Across both the Main (M) and Plants (P) tasks, *ReStraV* consistently performs near or above the baseline. On the Main (M) task, it demonstrates strong accuracies against AI generators (MuseV 93.52%, SVD 94.01%, CogVideo 93.52%, Mora 92.97%) and robust performance on natural videos (HD-VG/130M 91.07%), achieving a 93.01% average accuracy.

This robustness extends to the challenging Plants (P) task, where *ReStraV* obtains accuracies of 95.06% (MuseV), 97.83% (SVD), 92.38% (CogVideo), 91.24% (Mora), and 93.31% (HD-VG/130M), leading to a 93.96% average. This success may be attributed to *ReStraV*'s ability to capture specific curvature patterns inherent to "Plants" videos, which differ from more general artifacts. *ReStraV* remains highly effective as visualized by its position relative to the baseline spread (Fig. 13 in Appendix A.5).

Table 3: *ReStraV* vs. VideoSwin [39] fake video detection on VidProM[3]. "Seen generators" are those included in training; "Unseen generators" and "Future generators" were excluded from training. ↑ is better.

| Condition | | VideoSwin [39] | ReStraV (MLP) |
|---|---|---|---|
| **Seen generators** | Acc: 77.91 | **97.05** |
| | mAP: 75.33 | **98.78** |
| **Unseen generators** | Acc: 62.44 | **89.45** |
| **([62, 63])** | mAP: 59.61 | **97.32** |
| **Future generators** | Acc: 60.70 | **80.05** |
| **(Sora [68])** | mAP: 58.20 | **92.85** |

Table 4: Acc. (%) results of *ReStraV* vs. SoTA video based methods on the GenVidBench [17]. Table (a) shows results for the Main (M) task, and Table (b) for the Plants (P) task. ↑ is better.

**(a) GenVidBench - Main (M) Task Acc. (%)**

| Method | MuseV | SVD | CogVideo | Mora | HD-VG [Nat.] | Avg. |
|---|---|---|---|---|---|---|
| TSM [34] | 70.37 | 54.70 | 78.46 | 70.37 | 96.76 | 76.40 |
| X3D [37] | 92.39 | 37.27 | 65.72 | 49.60 | 97.51 | 77.09 |
| MVIT V2 [38] | 76.34 | **98.29** | 47.50 | **96.62** | 97.58 | 79.90 |
| SlowFast [36] | 12.25 | 12.68 | 38.34 | 45.93 | 93.63 | 41.66 |
| I3D [35] | 8.15 | 8.29 | 60.11 | 59.24 | 93.99 | 49.23 |
| VideoSwin [39] | 62.29 | 8.01 | 91.82 | 45.83 | **99.29** | 67.27 |
| **ReStraV** | **93.52** | 94.01 | **93.52** | 92.97 | 91.07 | **93.01** |

**(b) GenVidBench - Plants (P) Task Acc. (%)**

| Method | MuseV | SVD | CogVideo | Mora | HD-VG [Nat.] | Avg. |
|---|---|---|---|---|---|---|
| SlowFast [36] | 81.63 | 29.80 | 75.31 | 19.31 | 73.03 | 55.30 |
| I3D [35] | 39.18 | 23.27 | 91.98 | 78.38 | 78.42 | 62.15 |
| VideoSwin [39] | 57.96 | 7.35 | 92.59 | 47.88 | **98.76** | 52.86 |
| TPN [40] | 43.67 | 20.00 | 85.80 | 86.87 | 94.61 | 64.24 |
| UniFormer V2 [41] | 13.88 | 7.76 | 41.98 | **95.75** | 97.93 | 64.76 |
| TimeSformer [42] | 77.96 | 29.80 | **96.30** | 93.44 | 87.14 | 75.09 |
| **ReStraV** | **95.06** | **97.83** | 92.38 | 91.24 | 93.31 | **96.96** |

**C) One-to-many generalization Test.** We reproduced the one-to-many task from DeMamba [43], training on a single generator and testing on multiple unseen ones (Sora [68], MorphStudio [73], Gen2 [74], HotShot [75], Lavie [5], Show [76], MoonValley [77], Crafter [67], ModelScope [49] and WildScrape[43]). Table 5 shows average results across three training conditions. Notably, *ReStraV* achieves competitive or superior scores in most scenarios against specialized video detectors (TALL [78], NPR [79], STIL [80], and DeMamba [43]) using only the extracted trajectories.

**D) Zero-shot Generalization Test.** We tested zero-shot generalization on Google's Veo3 [65], a state-of-the-art model acclaimed for its ability to generate videos with plausible physical interactions and consistent object interactions (qualitative frame samples in Appendix A.6). Using only the MLP trained in Section 6 (without any Veo3 videos in training), we

Table 5: One-to-many: training on one generator, testing on unseen generators (Sora, MorphStudio, Gen2, HotShot, Lavie, Show-1, MoonValley, Crafter, ModelScope, WildScrape). Avarage results from DeMamba benchmark [43]. ↑ is better.

| Method | Train: Pika | | | Train: SEINE | | | Train: OpenSora | | |
|---|---|---|---|---|---|---|---|---|---|
| | R | F1 | AP | R | F1 | AP | R | F1 | AP |
| NPR | 0.514 | 0.531 | 0.650 | 0.462 | 0.539 | 0.611 | 0.593 | 0.523 | 0.576 |
| STIL | 0.738 | 0.517 | 0.630 | 0.724 | 0.506 | 0.608 | 0.434 | 0.489 | 0.526 |
| TALL | 0.714 | 0.557 | 0.623 | 0.657 | 0.609 | 0.681 | 0.492 | 0.532 | 0.571 |
| DeMamba | **0.757** | 0.726 | **0.817** | 0.810 | 0.787 | **0.894** | 0.738 | 0.671 | **0.738** |
| **ReStraV** | 0.735 | **0.827** | 0.797 | **0.820** | **0.898** | 0.854 | **0.771** | **0.797** | 0.717 |

tested on 200 Veo3 versus 200 natural video pairs and achieved 83.2% accuracy, 85.1% F1, and 86.9% AUROC. This shows that the curvature-based detection signal can generalizes to future generators not represented in the training distribution, supporting our hypothesis that current generative models fundamentally struggle to replicate the temporal smoothness characteristic of natural world dynamics in learned representation spaces.

**E) ReStraV vs. VLM detector on Physics-IQ dataset (matched real and generated videos).** As a third and perhaps the most challenging test, we assess *ReStraV* on *matched* pairs of natural and generated videos from the Physics-IQ dataset. This dataset consists of real-world physical interactions and is special in the sense that it consists of both natural and AI-generated videos (198 per source) that are based on the very same starting frame(s): identical scenes, identical objects, identical lighting conditions as described in Section 5. We report [26]'s evaluation using a two-alternative forced-choice (2AFC) paradigm (a gold-standard psychophysical protocol). In each trial, a model sees a pair of videos: one real, one AI-generated.

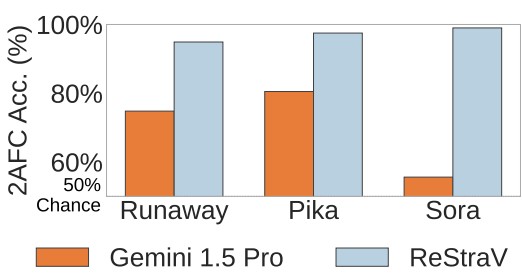

Figure 7: Fake video detection on Physics-IQ (*matched* real/generated video pairs). Gemini results from [26]. Despite the challenging task, ReStraV reliably identifies fake videos.

The task of identifying the AI-generated video is especially challenging due to the matched video nature. Motamed et al. [26] report results for a VLM, Gemini 1.5 Pro [66]. Gemini identifies Runway and Pika videos with reasonable accuracy (74.8% and 80.5% respectively), but Sora videos prove challenging (55.6%, near 50% chance) due to their photorealism.

We evaluate *ReStraV* in the same setting, comparing against reported numbers [26]. We compute mean aggregated video curvature (Eq. (1)) for each video and predict the one with higher mean curvature as "AI-generated." No further classifier training or calibration is performed. Fig. 7 shows the results: *ReStraV* attains 97.5% for Pika [46], 94.9% for Runway [81], and 99.0% for Sora. This near-perfect performance across all three generators demonstrates that simple curvature statistics robustly discriminate real from generated videos without model fine-tuning.

> **Takeaway 3:** *ReStraV* demonstrates robust generalization across diverse generators and OOD scenarios, showing neural representation trajectories (distance $d$ and curvature $\theta$) as an effective paradigm for AI video detection.

## 8 Discussion

**Summary.** As AI-generated videos look more and more realistic, it is increasingly important to develop methods that reliably detect AI-generated content. We here propose using simple statistics such as the angle between video frame representations, inspired by the perceptual straightening hypothesis from neuroscience [1], to distinguish natural from generated videos. The approach is compellingly simple, fast, cheap, and surprisingly effective: using a pre-trained feature space such as DinoV2, the resulting "fake video" signal reliably identifies generated videos with high accuracies, setting a new SoTA in fake video identification.

The surprising observation that natural videos have, on average, less curvature but at the same time a *higher* variance in their curvature demands attention. Prior work found that temporal transitions in natural videos latent representations follow highly sparse distributions [82]. That means most of the time there is very little change, but sometimes a large jump. In terms of curvature, this could mean that most of the time, natural videos follow a relatively straight line through representation space, but sometimes take a sharp term (perhaps a scene cut). Further investigation in trajectory geometry (e.g., curvature kurtosis) will help to shed light on this in future work.

**Implications for neuroscience.** The finding that natural videos trace *straighter* paths than AI-generated ones in a frozen vision transformer dovetails with the *perceptual straightening* phenomenon reported in perceptual decision tasks and brain recordings [1, 2]. In the brain, such straightening is often interpreted as a by-product of predictive coding: when the visual system internalizes the physical regularities of its environment, successive latent states become easier to extrapolate, reducing curvature in representation space. Our results imply that even task-agnostic, self-supervised networks acquire a comparable inductive bias—suggesting a shared computational pressure, across biological and artificial systems, to encode "intuitive physics" in a geometry that favors smooth trajectories [83].

Naturalistic straightening offers a concrete, quantitative handle for probing world-model formation in neural populations. Future work could ask whether curvature statistics in cortical population codes track the degree of physical realism in controlled stimuli, or whether manipulations that disrupt intuitive physics (e.g., gravity-defying motion) elicit the same curvature inflation we observe in synthetic videos. Such experiments would clarify whether the brain genuinely leverages trajectory geometry as an error-monitoring signal and how this relates to theories of disentangled, factorized latent representations of dynamics.

Our method is invariant to playing a video backwards. This is clearly unnatural, if things, e.g., fall up instead of down; though at the same time this also would not be an instance of an AI-generated video. The arrow of time [84–86] can be a strong signal, but our metric is invariant to a reversal of time.

**Limitations.** Goodhart's law states that *"when a measure becomes a target, it ceases to be a good measure"*. Likewise, in the context of fake video detection, it is conceivable that someone developing a video model could train it in a way that optimizes for deceiving detection measures. This concern generally applies to all public detection methods, including ours. As a possible mitigation strategy, it may be helpful to employ several detection methods in tandem, since it may be harder to game multiple metrics simultaneously without sacrificing video quality. Furthermore, as video models become more and more capable of generating realistic, natural-looking videos, it is possible that future video models may not show the same statistical discrepancies between real and generated videos anymore, though this is hard to predict in advance.

**Broader Impacts** AI-generated video increasingly fuels fake news and disinformation [15]. *ReStraV* aims to positively impact this by enhancing content authentication. With AI-driven fraud like deepfake scams reportedly surging (e.g., a reported 2137% rise in financial sector attempts over three years [87]), efficient detection methods like *ReStraV* are becoming fundamental.

However, deploying detection technologies like *ReStraV* faces an "arms race" with evolving generation methods (see Limitations; also [22]). Additionally, biases inherited from pre-trained encoders (e.g., DINOv2 [25]) may cause fairness issues across diverse content [88]. Mitigating these risks demands ongoing research, transparency about limitations, and using detectors primarily to aid human judgment. Key strategies include careful contextual deployment, rigorous bias auditing and debiasing efforts [88], promoting media literacy [15], and advancing complementary methods like robust content watermarking [20].

The importance of AI-safety measures [89] is increasingly reflected in policy initiatives like the Coalition for Content Provenance and Authenticity [90] and the EU AI Act [91], both vital for a trustworthy digital ecosystem. However, deploying detection tools at scale presents its own challenges, especially concerning user privacy. To address this, detectors can be distributed and trained using privacy-by-design principles [92–94]. Within this framework, tools like *ReStraV* are crucial for ensuring the digital ecosystem remains grounded in reality, providing a critical defense against the long-term risk of epistemic decay in world models [95, 96].

## Acknowledgements

This research was partly funded by Honda Research Institute Europe and Cold Spring Harbor Laboratory. We would like to thank Eero Simoncelli for insightful discussions and feedback, as well as Habon Issa, Filip Vercruysse, CiCi Xingyu Zheng, Alexei Koulakov, Andrea Castellani, Sebastian Schmitt, Andrea Moleri, Xavier Bonet-Monroig, Linus Ekstrøm, Riza Velioglu, Riccardo Cadei, and Christopher Van Buren for their helpful suggestions during the preparation of this manuscript, and Elena Ziegler for the appendix art illustration.

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

# Appendix Contents

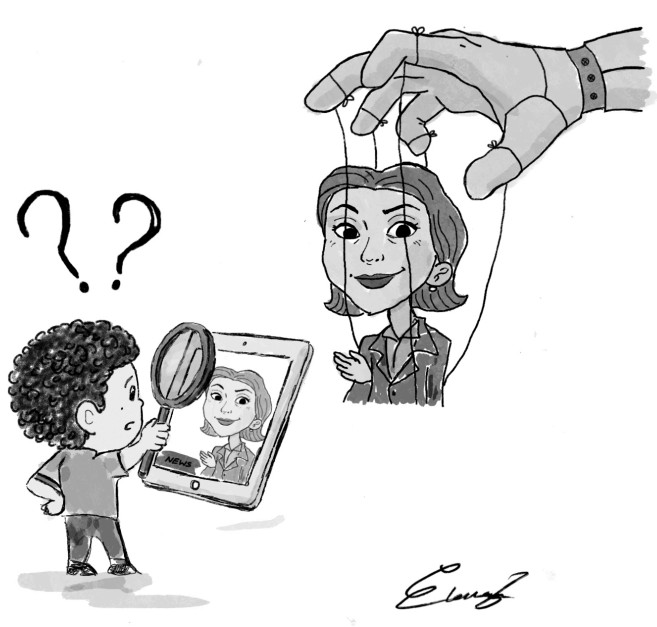

# A AI Video Generation

Early video generation used deep generative models like GANs and variational methods. [97] used VGANs for tiny video loops; [7] introduced MoCoGAN to separate motion and content. These pioneering methods, despite enabling synthetic video generation, often produced blurry or temporally incoherent results. [98] addressed future frame uncertainty with the Stochastic Variational Video Prediction (SV2P) model, using stochastic latent variables for diverse video sequences.

Diffusion models marked a significant breakthrough. Foundational methods like DDPM [12] (images) and Latent Diffusion [99] achieved high-fidelity generation via iterative denoising. Video Diffusion Models (VDMs) then addressed temporal consistency, e.g., using time-conditioned 3D U-Nets [100]. This led to prominent text-to-video systems like Imagen Video [100] and Make-A-Video [48], often using cascaded super-resolution. Latent diffusion variants like Text2Video-Zero [63] and ModelScope [3] improved efficiency by operating in latent spaces. Generative foundation models have diversified beyond diffusion. OpenAI's Sora [64] showed strong text-to-video capabilities using transformer decoders. Runway Gen-3 [81] uses autoregressive generation for temporal dynamics; Pika [46] combines diffusion and autoregressive decoding for improved coherence and quality.

Despite these advances, robust temporal consistency and physical plausibility remain significant challenges. Temporal inconsistencies occur even in sophisticated models like Stable Video Diffusion (SVD) [70, 101]. Even top models like Sora [64] and Google's VideoPoet [102] show coherence issues or generate implausible scenarios [26]. A critical gap is the lack of explicit physical dynamics modeling and coherent scene understanding, leading to unrealistic motion and interactions [26]. Incorporating world models to learn physical principles and causality [95, 103] is a promising research direction. These could mitigate temporal inconsistencies by enforcing structured scene understanding and dynamic constraints [103, 104]. Motivated by these persistent limitations in achieving temporal and *physical* realism, our work proposes detection methods that exploit subtle irregularities in the geometric properties of neural representations [1, 2].

## A.1 Qualitative Examples of Perceptual Trajectories

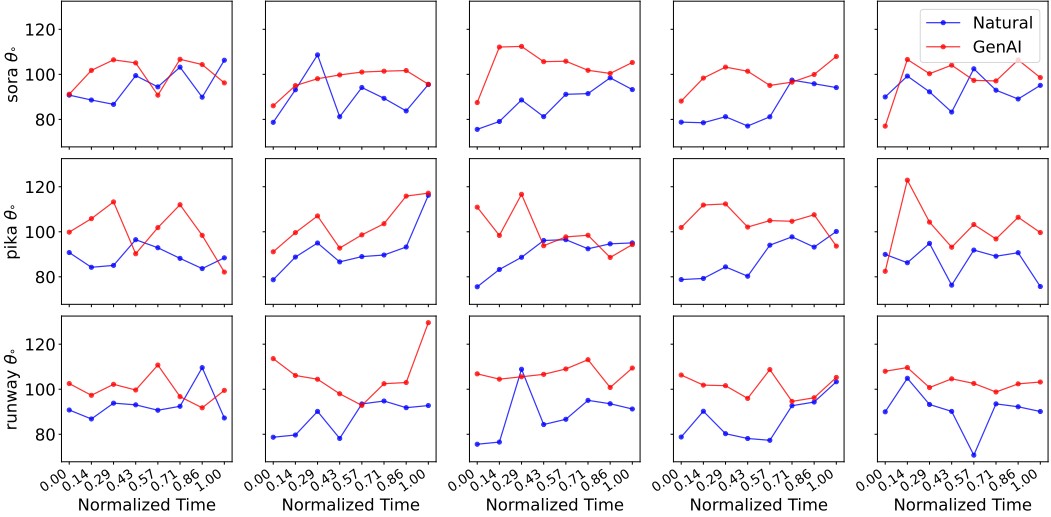

Figure 8: **Examples of raw curvature trajectories** ($\theta_i$) over normalized time for pairs of natural videos (blue) and AI-generated videos (red) from different generative models (Sora, Pika, Runway). Each column represents a different example pair. These qualitative examples illustrate the tendency for AI-generated videos to exhibit different curvature patterns, with more erratic fluctuations compared to their natural counterparts when viewed in the DINOv2 representation space.

To provide a more intuitive understanding of how curvature trajectories differ, Figure 8 presents several qualitative examples. Each plot shows the sequence of calculated curvature values ($\theta_i$) for a natural video (blue line) and a corresponding AI-generated video (red line) from the Sora [68], Pika [46], or Runway [81] models, after processing through the DINOv2 encoder as in Section 5.

While individual trajectories can be noisy, these examples visually highlight common tendencies observed: AI-generated videos frequently display trajectories with different overall levels of curvature, more pronounced peaks, or more erratic behavior compared to the often smoother or distinctly patterned trajectories of natural videos.

## A.2 ReStraV Classifiers Results Analysis

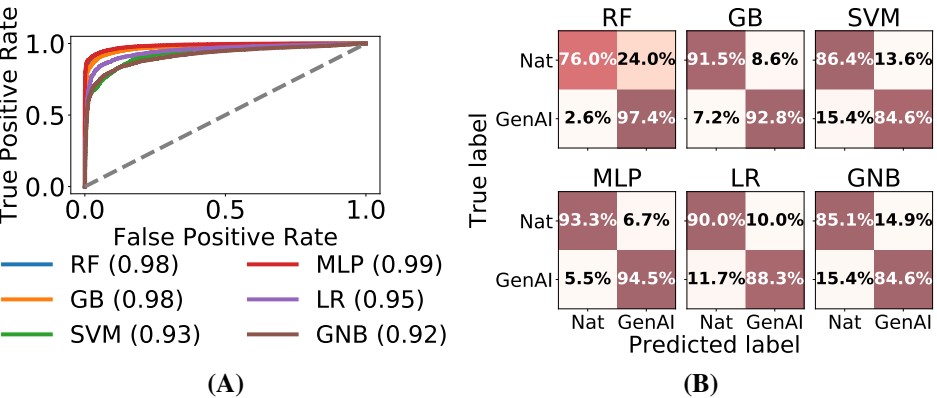

Figure 9: **(A) ROC curves** for various classifiers (Logistic Regression, Gaussian Naive Bayes, Random Forest, Gradient Boosting, SVM, MLP) on the test set. The MLP achieve the highest AUROC. **(B) Normalized confusion matrix for the ReStraV classifiers** on the test set, illustrating rates for both natural (Nat) and AI-generated (GenAI) classes. Values are percentages.

Figure 9A presents the ROC curves for all classifiers detailed in Table 1. The curve for the MLP (ReStraV) is closest to the top-left corner and with the largest area, AUROC (98.63%). Figure 9B displays the normalized confusion matrices for the *ReStraV* classifiers from Section 6. The strong diagonal elements (e.g., correctly identifying natural videos as "Nat" and AI-generated as "GenAI") and low off-diagonal values highlight the effectiveness. Specifically, correctly classifies a high percentage of both natural and AI-generated instances, aligning with the balanced accuracy and individual precision/recall/specificity metrics reported in Table 1.

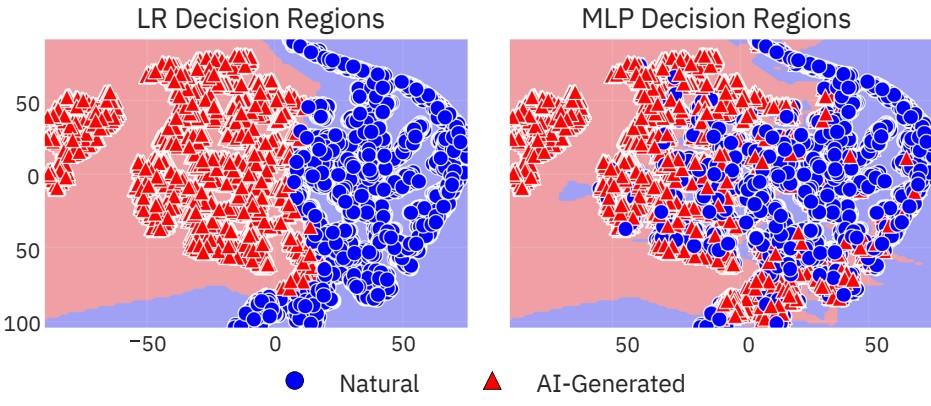

Figure 10: **Decision boundaries for Logistic Regression (LR) and Multi-Layer Perceptron (MLP)** *ResTraV***'s classifiers.** The plots illustrate how these models partition a 2D projection of the feature space (detailed in Section 6) to distinguish between natural (blue circles) and AI-generated (red triangles) videos. This comparison highlights the different decision boundaries learned by a linear (LR) and a non-linear (MLP) model when applied to the geometric trajectory features.

For demonstration purpose, we construct Voronoi tessellations to visualize the decision boundaries obtained from two different classification models: Logistic Regression (LR) and a Multi Layer Preceptor (MLP) classifier from Section 6 in Fig. 10. The visualization underscores the benefit of

using a non-linear classifier for this task, given the nature of the feature space derived from *ReStraV*'s geometric analysis.

## A.3 Feature Importance Analysis

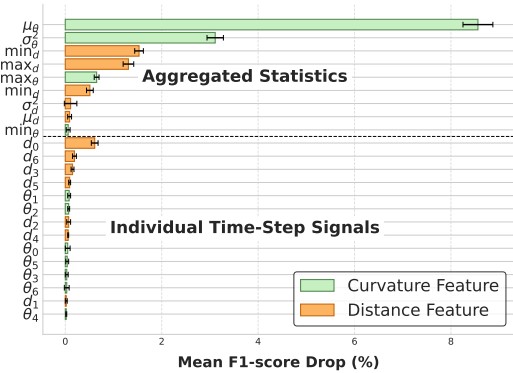

Figure 11: **Permutation Feature Importance.** The plot shows the mean drop in F1-score (%) when each feature is permuted, with error bars indicating standard deviation. Features are grouped into "Aggregated Statistics" (e.g., mean, variance) and "Individual Time-Step Signals" (e.g., $d_0$, $\theta_1$). The results highlight the overwhelming importance of the mean curvature ($\mu_\theta$) and other aggregated statistics in distinguishing natural from AI-generated videos.

To identify which of the 21 features contributed most to the performance of our best classifier (the MLP), we conducted a permutation feature importance analysis. We trained the MLP on the 50,000 AI-generated and 50,000 natural videos described in Section 5 and then measured the drop in F1-score when each feature was individually shuffled. A larger drop indicates a more important feature. The results are visualized in Fig. 11.

The mean curvature ($\mu_\theta$) is unequivocally the most critical feature. Its importance is more than double that of the next most influential feature, the curvature variance ($\sigma_\theta^2$). This provides strong quantitative evidence that the overall "straightness" of a trajectory is the primary signal our method leverages. The eight aggregated statistical features occupy the top tiers of importance. In contrast, the features representing individual, time-specific distance and curvature values ($d_i$, $\theta_i$) have a much smaller impact, suggesting they provide complementary but less critical information. These findings validate the distributions shown in Figure 5, where the aggregated statistics for curvature and distance showed the clearest separation between natural and AI-generated videos. The classifier effectively learns to exploit these high-level geometric properties.

## A.4 Qualitative Examples of "Plants" Task

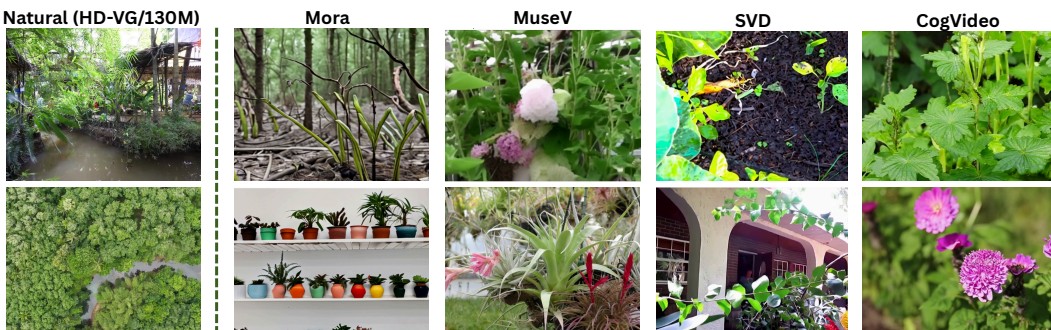

Figure 12: **Sample from the GenVidBench "Plants" task** [17] for natural video (HD-VG/130M) and AI-generated plant video frames (MuseV [69], SVD [70], CogVideo [71], and Mora [72]).

Figure 12 shows qualitive samples of "Plants" task (P) [17]. This task involves videos where the primary subject matter is various types of flora. Natural videos (HD-VG/130M [100]) exhibit typical characteristics of real-world plant footage. The AI-generated examples from MuseV [69], SVD [70], CogVideo [71], and Mora [72] showcase the capabilities of these models in synthesizing plant-related content. While visually plausible, these AI-generated videos contain subtle temporal unnatural patterns (e.g., texture evolution) that *ReStraV* detect through its geometric trajectory analysis. The performance of ReStraV on this hard task are described in Table 4(P).

## A.5   Results Distributions for Main Task (M) and Plants Task (P)

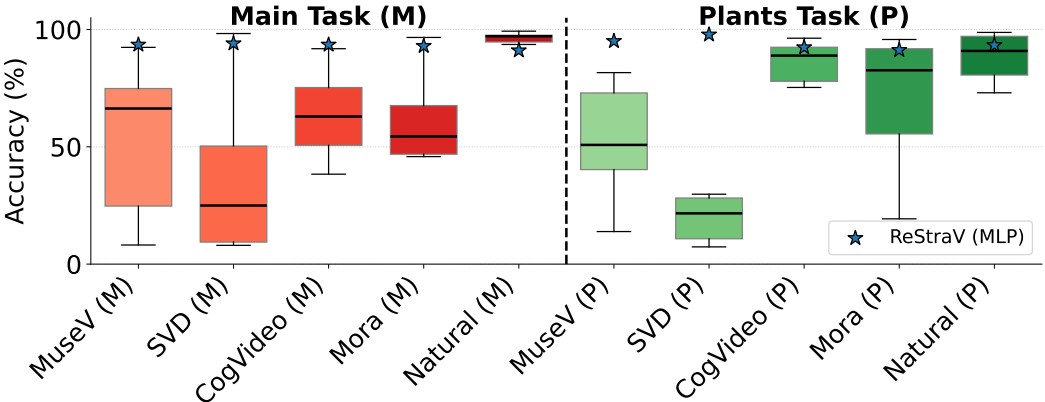

Figure 13: Accuracy distributions of *ReStraV* (MLP) and SoTA methods on GenVidBench [17]. Box-plots summarize performance on the Main task (red distributions) and Plants task (green distributions) across the different generative models within each task.

Figure 13 visualizes the accuracy distributions of various detectors, including *ReStraV* (MLP), on the GenVidBench Main (red distributions) and Plants (green distributions) tasks. The boxplots illustrate the median accuracy, interquartile range (IQR), and overall spread of performance for each method across the different generative models within each task. *ReStraV* (MLP) is consistently positioned at the higher end of the accuracy spectrum for both tasks, indicating more stable performance across different generators. For the Main task, ReStraV's median and overall distribution are visibly superior. For the Plants task, ReStraV again demonstrates leading performances. This visualization complements Table 4 by providing a overview of performance consistency and superiority.

## A.6   Frame Samples from the Veo3 Model for Zero-shot Generalization Test

To provide a qualitative sense of the videos used in our zero-shot generalization test (Section 7, Paragraph D), Figure 14 presents sample frames generated by the Veo3 model. These examples illustrate the high quality of the content our method was tested against without any prior training on this specific generator.

# B   Ablation Studies

We performed ablation studies to understand the impact of key frame sampling parameters on the performance and efficiency of *ReStraV*. We randomly selected 10,000 AI-generated videos by Sora [68] from VidProM [3] and 10,000 natural videos from DVSC2023 [47]. The Sora [68] videos, with longer lengths (5s) and high frame rate (30 FPS as the natural videos), provide a robust basis for evaluating a wide range of sampling parameters, making them a good representative case for the AI-generated set. Performance is evaluated using Accuracy (%), AUROC (%), and F 1 Score (%), with inference time measured in milliseconds (ms). The shaded regions in the plots represent ±1 standard deviation around the mean, based on multiple runs involving different random video samples and 50/50 train-test partitioning. We use the best performer classifier (two-layer MLP $(64 \rightarrow 32)$) from Section 6 of the main paper.

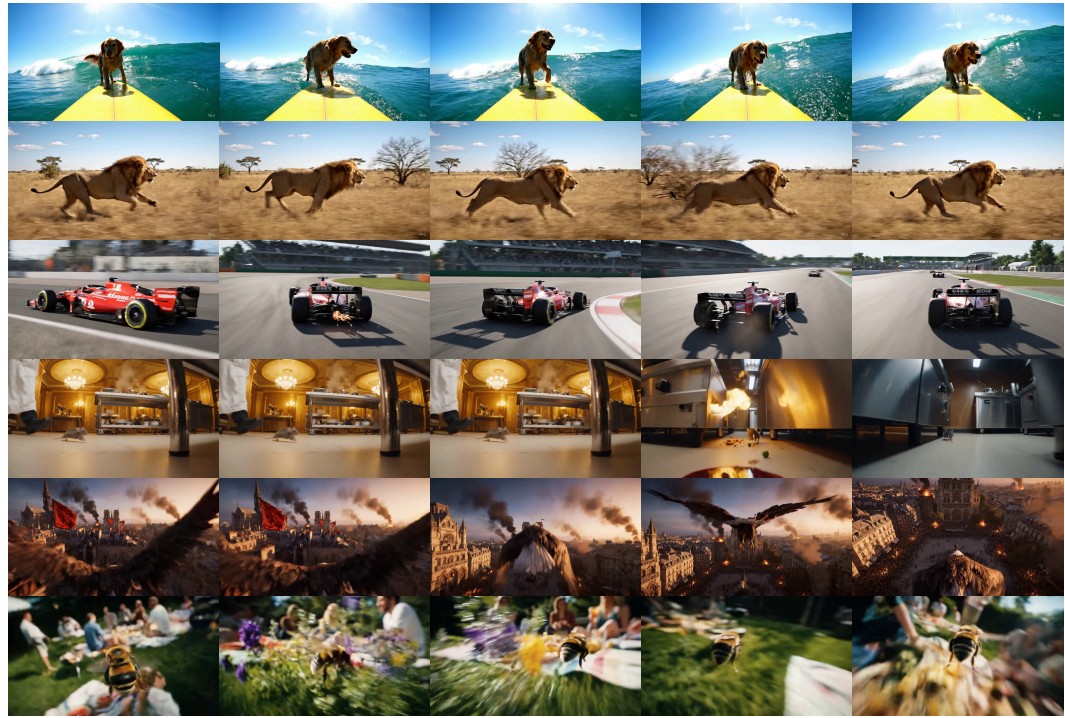

Figure 14: **Qualitative frame examples from the Veo3 [65] model.** These images showcase the high-fidelity content used for the zero-shot generalization test.

## B.1 Impact of Video Length and Sampling Density

We investigated how the length of the video analyzed and the density of frame sampling within a fixed window affect *ReStraV*'s performance. The results are shown in Supplementary Figure 15.

**Panel (a)** of Supplementary Figure 15 illustrates the effect of varying the analyzed video length from 1 to 5 seconds, from which *ReStraV* processes a 2-second segment by sampling every 3rd frame (at 30 FPS). This results in the number of frames ($T$) input to DINOv2 [25] being 10, 20, 30, 40, and 50 for the respective conditions. As shown, performance metrics (Accuracy, AUROC, and $F_1$ score for AI-generated content) improve as the analyzed video length increases, with AUROC exceeding 96% for 2-second segments ($T \approx 20$) and reaching approximately 98% for 5-second segments ($T = 50$). Inference time increases linearly with $T$. The 2-second segment analysis, as used in our main paper (where $T = 24$ frames are processed), offers a strong balance between high performance and computational efficiency (observed around 40-48ms in related tests).

**Panels (b) and (c) of** Supplementary Figure 15 show the sampling density within a fixed 2-second source video (30 FPS, 60 total frames). Panel (b) shows performance against the sampling interval $k$ (where every $k^{th}$ frame is taken). The results indicate optimal performance when $k = 3$ ($T = 20$ frames), achieving an AUROC of $\approx 97\%$. Performance degrades for sparser sampling (e.g., $k = 5$, $T = 12$) and also for very dense sampling (e.g., $k = 1$, $T = 60$). Panel (c) plots performance directly against the number of sampled frames $T$, confirming peak performance at $T = 20$.

## B.2 Robustness to Temporal Window Position

We also studied the impact of the starting position of the 2-second window when applied to 5s videos. A 2-second window (processed with *ReStraV*'s standard $T = 24$ frames) was slid across a 5-second video with a step of 10 frames (approximately 0.33s at 30 FPS).

As shown in Supplementary Figure 16, the detection performance remains largely robust regardless of the window's start time. A slight U-shaped trend is observed, with marginally higher performance at the beginning and end of the analyzed 0-3s window start time range, and a minor dip when the

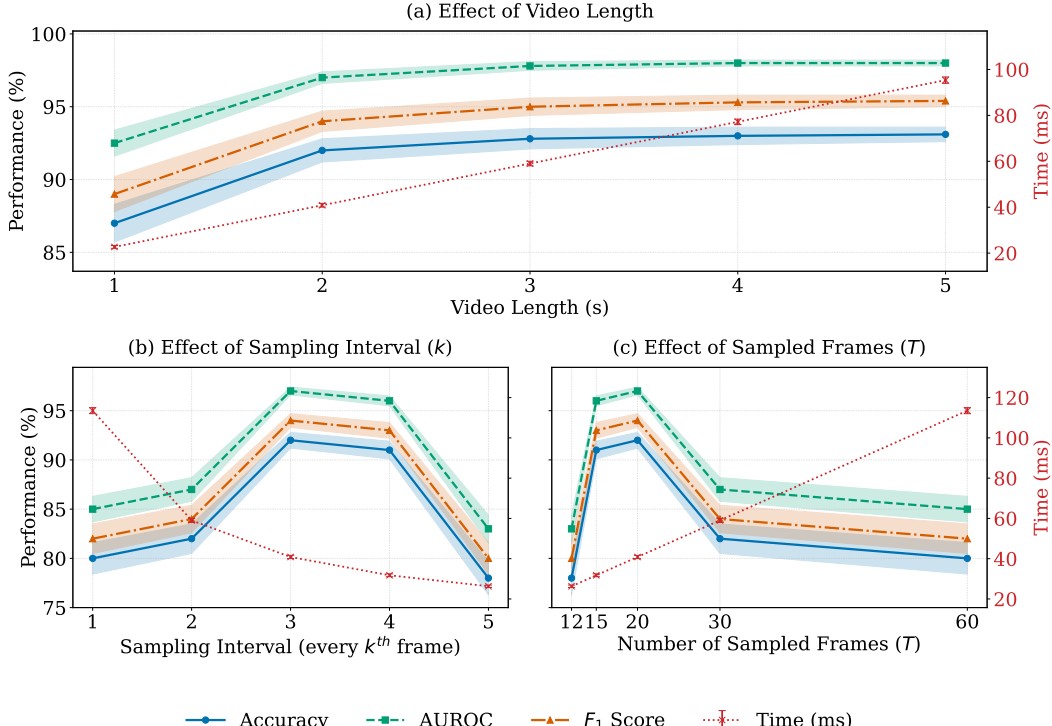

Figure 15: **Ablation on Video Length, Sampling Interval, and Number of Frames.** (a) Effect of analyzed video length. (b) Effect of sampling interval ($k$) for a 2s video. (c) Effect of total sampled frames ($T$) for a 2s video, derived from varying $k$.

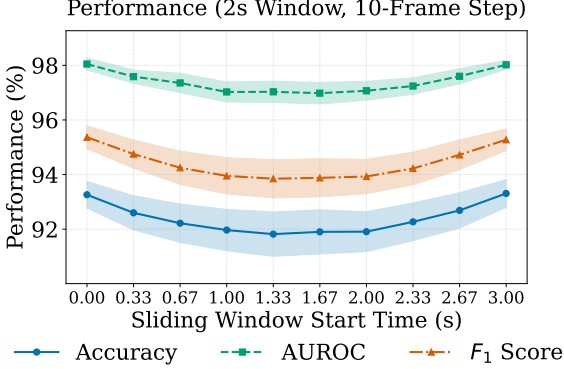

Figure 16: **Ablation Study on Sliding Window Start Time.** Performance of *ReStraV* when a 2s window (with $T = 24$ frames) slides across a 5s video with a 10-frame step.

window is centered. This demonstrates that ReStraV is not overly sensitive to the precise temporal segment analyzed within a longer video.

## C Analysis of Scene Cut Frequency

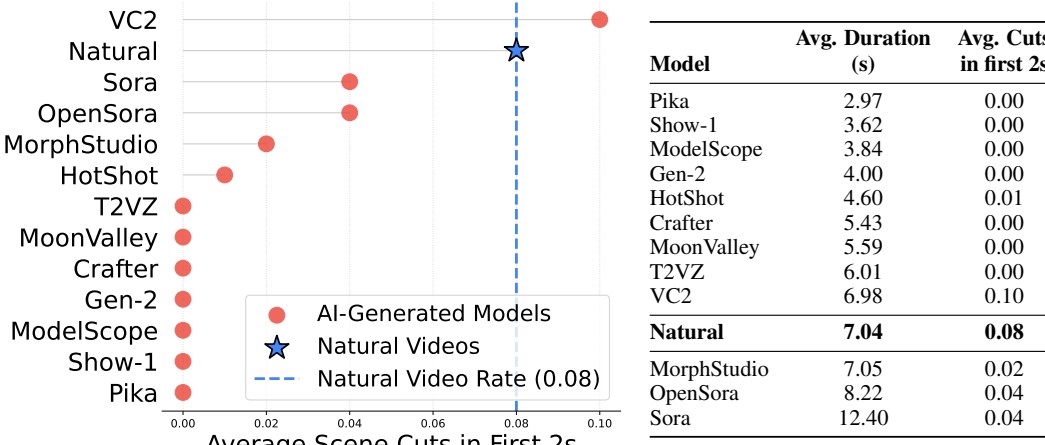

| Model | Avg. Duration (s) | Avg. Cuts in first 2s |
|---|---|---|
| Pika | 2.97 | 0.00 |
| Show-1 | 3.62 | 0.00 |
| ModelScope | 3.84 | 0.00 |
| Gen-2 | 4.00 | 0.00 |
| HotShot | 4.60 | 0.01 |
| Crafter | 5.43 | 0.00 |
| MoonValley | 5.59 | 0.00 |
| T2VZ | 6.01 | 0.00 |
| VC2 | 6.98 | 0.10 |
| **Natural** | **7.04** | **0.08** |
| MorphStudio | 7.05 | 0.02 |
| OpenSora | 8.22 | 0.04 |
| Sora | 12.40 | 0.04 |

Figure 17: **Analysis of Scene Cut Frequency.** The lollipop plot (left) and table (right) show the average number of hard scene cuts within the first 2s for each video source. The dashed blue line in the plot indicates the rate for natural videos. Both visualizations confirm that the scene cut frequency is low across all models and not a significant confounding factor.

An important consideration for AI-generated video detection is the effect of hard scene cuts, as the straightening hypothesis is not expected to hold across shot boundaries. This presents a potential confound: if AI-generated videos simply contained a higher frequency of scene cuts, it could partly explain our classifier's performance.

To investigate this possibility, we analyzed 13,000 AI-generated and 13,000 natural videos using the "scenedetect" [105] library. The results, presented in Fig. 17, show that the average number of scene cuts is low and comparable for both natural and AI-generated videos. The plot (left) visualizes this comparison, showing all models clustering near the natural video baseline (dashed line), while the table (right) provides the precise data.

Our method's reliance on a short, 2s analysis window inherently reduces the probability of encountering a scene cut. The overall robustness of our approach is further confirmed by the ablation study in Appendix B.2, which shows stable detection performance regardless of the analysis window's temporal position.

## D Computational Environment

All experiments presented in this paper were conducted on a system equipped with NVIDIA RTX-2080 GPUs, each with 8GB of VRAM. The feature extraction process using the DINOv2 ViT-S/14 model, which involves a forward pass for an 24-frame batch, takes approximately 43.6 milliseconds.

## E Dataset Licenses and Sources

- **VidProM [3]:** This dataset was employed for training our video classifier (Section 6) and for benchmarking in Section 7 and Section 7. The VidProM dataset is offered for non-commercial research purposes under CC BY-NC 4.0 license.

- **GenVidBench [17]:** GenVidBench was used for benchmarking in Section 7). The GenVidBench dataset and its associated code are under CC BY-NC-SA 4.0 license.

- **Physics-IQ [26]:** This dataset facilitated the evaluation of our method on matched pairs of natural and AI-generated videos that depict physical interactions (Sec. 4, Sec. 7). The Physics-IQ dataset is available under the Apache License 2.0.
- **DVSC2023 (Natural Video Source for Classifier Training):** As explained in Section 5, a set of 50,000 natural videos for training (Section 6) was sourced from DVSC2023 [47]. DVSC2023 is under the CC BY-SA 4.0 llicense.

