# OpenReview forum: "AI-Generated Video Detection via Perceptual Straightening"
_NeurIPS.cc/2025/Conference — NeurIPS 2025 poster_

### Official Review · Reviewer_NU9X · 2025-06-30

**Clarity:** 3
**Significance:** 3
**Originality:** 3
**Rating:** 5
**Confidence:** 3

**Summary:**

Grounded on the perceptual straightening hypothesis, this paper presents an innovative approach to distinguish between natural videos and AI-generated videos by leveraging the distance and curvature between frame features. The simplicity and effectiveness of this method are commendable, and its relatively low computational cost is also a significant advantage. By solely utilizing statistical metrics in the perceptual straightening space to train classifiers, the authors have achieved remarkable results, which offers a fresh and valuable perspective on the detection of AIGC-detection.

**Questions:**

1. Compared to RF, what are the advantages of MLP? Is the computational time cost of RF measured on the CPU?
2. Could the differences detected here be due to noise interference? If AI-generated videos are augmented with simulated noise or compression artifacts, would the algorithm still be able to distinguish them effectively?
3. How many input frames are acceptable for this algorithm and the compared algorithms? Is it because we input more frames that we have a stronger ability to distinguish?

**Ethical Concerns:**

["NO or VERY MINOR ethics concerns only"]

**Final Justification:**

I appreciate the authors' original motivation to distinguish between trajectory patterns in natural and AI-generated videos, as well as their efforts to clarify the details and additional comparisons suggested by other reviewers. I am happy to keep my orginal positive score.

**Limitations:**

Yes, it mentioned that developers of video models might optimize them in a way that deceives detection measures. However, it is necessary to propose advanced detectors. We should be more capable of distinguishing between real and fake videos than those who create AI-generated videos.

**Quality:**

3

**Strengths And Weaknesses:**

Strengths:
1.	Quality: The paper presents a well-constructed hypothesis regarding the trajectory patterns in natural versus AI-generated videos. The authors provide substantial evidence to support their claims, demonstrating thorough research and analysis. The detection pipeline they developed, utilizing a standard ViT and lightweight classifiers, appears to be robust and effective, as evidenced by the significant results across multiple benchmarks. The quality of the research is further enhanced by the authors' meticulous attention to detail in highlighting the distinctions between the two types of videos.
2.	Clarity: The use of factual evidence to illustrate the differences between natural and AI-generated videos is particularly effective in enhancing clarity.
3.	Significance: The findings that natural videos tend to have smoother and more consistent trajectories compared to the irregular transitions in AI-generated videos offer valuable insights. The significant results across numerous benchmarks further highlight the practical importance of this research.
4.	Originality: The paper introduces a novel perspective on detecting AI-generated videos by focusing on perceptual straightening and trajectory patterns. This approach is innovative and represents a departure from traditional methods.
Weaknesses:
In line622, it mentions that Fig 10 shows the differences between RF and LR, but the caption of the figure states that it illustrates the differences between MLP and LR. Which one is correct?

---

> ### Author Rebuttal · Authors · 2025-07-31
>
> Thank you very much for the positive evaluation of our work *(“simplicity and effectiveness”, “remarkable results”, “fresh and valuable perspective*”) as well as the feedback and clarifying questions.
>
> *“In line622, it mentions that Fig 10 shows the differences between RF and LR, but the caption of the figure states that it illustrates the differences between MLP and LR. Which one is correct?”:* Thank you for catching the discrepancy in the appendix. We will fix line 622 to make it clear that Figure 10 is about LR and MLP in the final version of the paper.
>
> *“Compared to RF, what are the advantages of MLP? Is the computational time cost of RF measured on the CPU?”*: Regarding the comparison between RF and MLP, as detailed in Table 1, while RF has a marginally higher accuracy (94.24% vs. 94.17%) and F1-score (96.53% vs. 96.48%), our simple two-layer MLP (64→32  with a native implementation in sklearn) achieves a higher AUROC (98.63% vs. 98.03%). This indicates the MLP provides a better trade-off between true positives and false positives across different thresholds, making it a more robust classifier overall. The inference time for the entire pipeline (DINOv2 feature extraction on a GPU \+ classifier on CPU) is nearly identical for both (RF: 48.14 ms, MLP: 48.12 ms). The classifier portion itself is a minimal component of this total time.
>
> *“If AI-generated videos are augmented with simulated noise or compression artifacts, would the algorithm still be able to distinguish them effectively?”:* This is an excellent question regarding the method's robustness. Our core hypothesis is that AI-generated videos exhibit more curved, irregular trajectories in the DINOv2 representation space compared to the straighter paths of natural videos. Adding uncorrelated noise or compression artifacts would introduce high-frequency, erratic changes between frames. In the representation space, this would almost certainly make a video's trajectory even more curved. Therefore, augmenting an AI-generated video with such noise would likely increase its detected curvature, making it more distinct from a natural video and thus easier for our method to classify.
>
> *“How many input frames are acceptable for this algorithm and the compared algorithms? Is it because we input more frames that we have a stronger ability to distinguish?”:* Thank you for the question, this is a key aspect of our method's efficiency. Please refer to our ablation study in "Supplementary Material: zip" Figure 1 and 2\. For a fixed 2-second video clip, the best performance is achieved with \~20-24 sampled frames. Using significantly fewer frames fails to capture enough temporal information, while using too many (e.g., 60 frames from the same 2s clip) actually degrades performance, likely by introducing redundant, high-frequency signals. Our choice of 24 frames with a sampling interval of 3 (8 final frames in total) is therefore a well-justified trade-off between accuracy and efficiency. When analyzing longer videos, performance generally improves. We observed that accuracy and AUROC increase as the source video length goes from 1 to 5 seconds, though with diminishing returns after the 2-3 second mark.

---

> > ### Comment · Reviewer_NU9X · 2025-08-07
> >
> > The reviewer thanks the authors for a detailed response, which mostly addresses my concern. I will keep the rate unchanged.

---

> ### Author Response · Authors · 2025-08-07
>
> Thank you again for your feedback and support regarding our work!

---

### Official Review · Reviewer_YPpG · 2025-07-02

**Clarity:** 3
**Significance:** 3
**Originality:** 2
**Rating:** 4
**Confidence:** 3

**Summary:**

This paper proposes ReStraV, a novel method for detecting AI-generated videos by analyzing temporal curvature and distance in neural representations. Inspired by the "perceptual linearization" hypothesis. ReStraV uses the pretrained DINOv2 visual transformer to quantify deviations in video trajectories. It shows that AI-generated videos have distinct curvature patterns compared to real videos. The method achieves state-of-the-art performance, with 97.17% accuracy and 98.63% AUROC on the VidProM benchmark, offering an efficient, low-cost solution for AI video detection.

**Questions:**

See weaknesses

**Ethical Concerns:**

["NO or VERY MINOR ethics concerns only"]

**Final Justification:**

On the positive side, the paper gives a novel approach and obtains strong empirical results. On the negative side, the method lacks certain theoretical principled guarantee. I will retain my original positive score.

**Limitations:**

Yes

**Quality:**

3

**Strengths And Weaknesses:**

Strengths
1. Clarity of Writing: The paper is clearly written and accessible, making it easy to follow the motivation and methodology.
2. Strong Empirical Results: The method shows significant improvements in detection performance, with accuracy increasing from 50–80% to 90–99% on benchmark datasets (e.g., Table 2).
3. Motivated by Prior Perception Theory: The use of curvature and distance in representation space is well-motivated by perceptual neuroscience literature.

Weaknesses
1. Limited Evaluation Scope: The experiments only evaluate one commercial video generation model (Pika). The method should be tested on a broader range of state-of-the-art generators such as Sora, Gen-3, Luma, Kling, and Vidu to validate generalization.
2. Limited Novelty: The core insight of the paper (e.g., the left and middle parts of Figure 1) appears closely derived from prior work [1, 2].
3. Potential Vulnerability to Adversarial Adaptation: It remains unclear whether the proposed detection method can be circumvented if curvature and distance metrics are explicitly regularized during the training of generative models.

[1] Olivier J Hénaff, Robbe LT Goris, and Eero P Simoncelli. Perceptual straightening of natural videos. Nature neuroscience, 22(6):984–991, 2019.
[2] Olivier J Hénaff, Yoon Bai, Julie A Charlton, Ian Nauhaus, Eero P Simoncelli, and Robbe LT Goris. Primary visual cortex straightens natural video trajectories. Nature communications, 12 (1):5982, 2021.

---

> ### Author Rebuttal · Authors · 2025-07-31
>
> Thank you for your positive feedback on our paper's clarity (*“clearly written”*), strong empirical results (*“significant improvements”*), and motivating theory (*“well-motivated by perceptual neuroscience literature”*). We appreciate the opportunity to address your questions:
>
> *“The experiments only evaluate one commercial video generation model (Pika)”*:  We would like to clarify a potential misunderstanding regarding the benchmarking that we will also clarify in the paper. *We cover a total of 18 generative models across multiple datasets, beyond Pika*. In Table 2 we evaluated four generators (*Pika, VideoCrafter2, Text2Video-Zero, ModelScope*) and tested generalization to the advanced *Sora* model as a "future generator" in Table 3\. We benchmarked against four different modern generators (*MuseV, SVD, CogVideo, Mora*) in Table 4\. We used matched real/generated pairs from *Runway, Pika, and Sora* in Physics-IQ Dataset (Figure 7). Additionally, during the rebuttal phase we tested our method on videos from the new Veo3 model. And we have reproduced the One-to-Many Generalization Task from Table 4 of \[1\], testing on Sora, MorpStudio, Gen2, HotShot, Lavie, Show-, MoonValley, Crafter, ModelScope.
>
> *“The core insight of the paper (e.g., the left and middle parts of Figure 1\) appears closely derived from prior work \[1, 2\]”:* Thank you for the point about the novelty of the work. We are indeed deeply inspired by the foundational work on perceptual straightening. Our novel contribution is not the concept itself, but its successful repurposing and practical application to the distinct and modern problem of AI-generated video detection, where we obtain SOTA results. To the best of our knowledge, we are the first to apply ideas from the perceptual straightening hypothesis to this task, and we will revise our manuscript to make this clear
>
> *“It remains unclear whether the proposed detection method can be circumvented if curvature and distance metrics are explicitly regularized during the training of generative models”:* We agree that this is a very important point that equally applies to all published detection methods. Similar to the “arms race” between adversarial attacks and defenses, the field of AI-generated video detection might see a similar dynamic. We explicitly acknowledge this in our paper's Discussion section under "Limitations". In the field of adversarial attacks, publishing attacks and defenses ultimately led to the development of robust techniques like adversarial training; we are hopeful that publishing AI-generated video detectors will similarly raise the bar to inspire a future generation of AI video generation models with more physical realism.
>
> \[1\] Chen et al, DeMamba: AI-Generated Video Detection on Million-Scale GenVideo Benchmark, 2024

---

> > ### Comment · Reviewer_YPpG · 2025-08-02
> >
> > I appreciate the clarifications. I would be better if the authors could using experimental results to check whether the third weakness holds: "whether the proposed detection method can be circumvented if curvature and distance metrics are explicitly regularized during the training of generative models". I will retain my original score.

---

> ### Author Response · Authors · 2025-08-06
> **Regarding circumvention**
>
> Thanks for your response, we’re glad to hear that our clarifications were helpful!
>
> While we agree that adversarially training a foundation model against our metrics would be a fascinating new project, the computational and energy costs of training even a single state-of-the-art generator make this infeasible with academic resources.
>
> Furthermore, our new analysis suggests that such regularization may not be straightforward. We performed a detailed correlation analysis between curvatures and metrics from the recent Physics-IQ benchmark [1]. This analysis revealed a *weak-to-moderate negative correlations (Spearman's ρ) between curvature and key metrics of visual fidelity*, including Mean Squared Error (ρ=−0.3109) and Variance IOU (ρ=−0.1248).
> This suggests that an attempt to explicitly minimize the curvature during training could inadvertently increase the model's reconstruction error. While more sophisticated implementations of such a intervention might be developed, our results demonstrate that this may not be a straightforward challenge. This suggests that *circumventing our detection method would be a non-trivial implementation, rather than a simple adjustment*.
>
> Therefore, within the scope of this project, we embrace what we believe is the strongest and most practical alternative: testing a strong “future” model that is multiple steps ahead of our method in the “arms race”. Specifically, we are testing the zero-shot generalization of our method against a new, unseen, state-of-the-art model (Google's Veo3) known for its exceptional temporal and physical coherence.
> A simple linear regressor, trained only on the features and models described in Section 5 of our paper, achieves the following results on this challenging, unseen data:
> Accuracy: 83.2%
> AUROC: 86.9%
> These results on a next-generation model demonstrate that the geometric artifacts our method captures are not superficial patterns of specific generators but are likely consequences of the current generation process.
> We’re adding this result, as well as a discussion on the possibility of regularizing the curvature during training, to our manuscript.
>
> [1] Saman Motamed, Laura Culp, Kevin Swersky, Priyank Jaini, and Robert Geirhos. Do generative video models understand physical principles?, 2025.

---

### Official Review · Reviewer_98v6 · 2025-07-03

**Clarity:** 3
**Significance:** 3
**Originality:** 3
**Rating:** 5
**Confidence:** 4

**Summary:**

The paper proposes a perceptual straightening hypothesis-based approach to discriminate real videos from AI-generated ones. Specifically, statistics related to distance and curvature of video representations derived using an off-the-shelf SSL model are used as features to train a simple MLP. The performance is shown to be competitive on the chosen set of datasets against image-based, video-based, and MLLM-based baselines.

**Questions:**

- Have the authors considered using the bandpass filters and nonlinearities in [1] for generating video representations? This would not only justify the straightening hypothesis but would also provide an interesting comparison. Further, [1] and [24] show that off-the-shelf models do not innately straighten natural videos. Therefore, such an experiment would be illuminating.

- Can you please suggest a more systematic approach to choosing an SSL model? The current methodology is completely empirical and adhoc.

- Why have standard baselines not been considered in the performance comparison? Please explain.

**Ethical Concerns:**

["NO or VERY MINOR ethics concerns only"]

**Final Justification:**

The detailed responses and the clarifications have addressed the concerns and questions. Raising my score.

**Limitations:**

Yes

**Quality:**

3

**Strengths And Weaknesses:**

** Strengths**
- The proposed perceptual straightening approach to discriminate read videos from AI-generated ones is lightweight and delivers good performance on the tested datasets.

**Weaknesses**
- The premise of perceptual straightening in SSL models is not justified. The original work [1] demonstrates straightening using bandpass filter plus nonlinearity models for the initial layers of the HVS. Further, [24] shows that SSL models can benefit from the straightening constraint. However, the proposed work assumes that the latent space of an off-the-shelf SSL model inherently straightens representations of natural videos. Further, the evidence of straightness using tSNE plots in Figure 4 is difficult to comprehend due to clutter. Furthermore, and crucially, Figure 8 (in A.2) does not provide conclusive evidence.

- Importantly, a discussion on why the proposed method works is limited. Specifically, why do SOTA generators like Sora behave the way they do in the latent space of SSL models?

- Also, the choice of the SSL model is rather adhoc.

- The paper does not cite relavant works from the perceptual quality assessment literature. These works are relevant since they use the notion of "straightness" to discriminate pristine (or real) videos from distorted (or unnatural) ones. Given this, the novelty of the proposed paper is limited. Please see for example: https://ieeexplore.ieee.org/stamp/stamp.jsp?arnumber=10667010, https://ieeexplore.ieee.org/abstract/document/9633248.

- The results are compared against three different approaches - image-based, video-based, and MLLM-based. However, the comparison is missing relevant works such as TALL, NPR, STIL, DeMamba etc.

- Please do a grammar check.

---

> ### Author Rebuttal · Authors · 2025-07-31
>
> We sincerely thank the reviewer for the insightful review. The feedback helped us identify areas where we can strengthen our manuscript. We are glad the reviewer acknowledges that our method is "lightweight" and "delivers good performance".We have worked to address all raised points below.
>
> *“The premise of perceptual straightening in SSL models is not justified”***:** This may be a misunderstanding that we will also clarify in the final version of the paper. We do not *“assume that the latent space of an off-the-shelf SSL model inherently straightens representations of natural videos”*, but we test which one reveals the AI-generated ones instead. We systematically investigate whether this is the case empirically: 1\) We show in Figure 2B that among eight diverse encoders, DINOv2 is most effective at maximizing the curvature difference between real and AI-generated content. 2\) We demonstrate that simple geometric and statistical features of the curvature and distance trajectory, as shown in Figure 5, are highly discriminative. 3\) We show that these features, when fed into simple classifiers, create a powerful detector that achieves SOTA performance, outperforming complex, dedicated detectors and foundational models (Gemini 1.5) on challenging benchmarks \[1,2,3\] (Tables 3 & 4 and Figure 7).
>
> *“The evidence of straightness using tSNE plots in Figure 4 is difficult to comprehend due to clutter. Furthermore, and crucially, Figure 8 (in A.2) does not provide conclusive evidence”***:** We see your point about clarity. Figure 4 (t-SNE) is meant to illustrate the emergent *class separability* in DINOv2's space together with Figure 1A, not absolute straightness which is very hard to visualize in the high dimensional embedding space. Figure 8 shows the raw curvature signal *before* we extract the statistical features that lead to our SOTA classification results as qualitative examples–one can see the separation in curvature between natural and AI in most plots. Following your comments, we will draw a convex hull or a density contour around the Natural vs. AI clusters to provide a clear visual boundary and emphasize the separation our method leverages in Figure 4 (t-SNE).  We will improve clarity in the caption of both figures.
>
> *“The choice of the SSL model is rather adhoc”***:** We agree with you that a purely theoretical motivation for selecting a specific encoder is not yet established. Our approach was therefore a principled empirical one. As shown in Figure 2B, we systematically screened different encoders from diverse architectures , e.g., CNNs (AlexNet, VGG-16, ResNet-50, SIN-ResNet-50), Transformers (CLIP,DINOv2), and training objectives, e.g., SSL (SimCLR-R50, BYOL-R50). We selected DINOv2 because it empirically demonstrated the largest discriminative power. In the absence of an *a priori* theory, this systematic empirical procedure appeared to be the most principled approach that avoids any unjustified or *ad hoc* choices. *We are very interested in any theories, suggestions or references that could explain this outcome, and we would be eager to incorporate them.*
>
> *“The paper does not cite relevant works from the perceptual quality assessment literature”*: Thank you for highlighting the relevant literature on VQA. This is a great suggestion\! The main difference is that the cited VQA papers address the problem of scoring perceptual quality, while ours focuses on AI-generated video detection. *Given the high perceptual quality of the latest AI-generated videos, these two tasks may be quite distinct.* We will gladly cite and discuss these related works in the revised manuscript.
>
> *“the comparison is missing relevant works such as TALL, NPR, STIL, DeMamba etc”*: We referred to the SOTA baselines in recent benchmark papers \[1,2,3\] for general AI-generated video detection. We included eight image based detectors (Table 3\) and nine general video detectors (Table 4\) and a foundational model (Gemini 1.5, Figure 7). Thank you for suggesting additional baselines. We agree that it would improve our paper to extend our test with the applicable ones. TALL, NPR and STIL are highly specialized for *deepfake face detection*, and thus cannot be used on the more general datasets we analyze. DeMamba \[4\] is a general detector; however, the model weights are not accessible at this point according to the author’s github (see issue \#16: “Due to the company's open source policy, we are temporarily unable to provide open source weights”). Nevertheless, the authors provide the full dataset to the public. We therefore reproduced the challenging “one-to-many generalization task” experiment of Table 4 in DeMamba \[4\]. The one-to-many generalization task involves training on one baseline category and then testing on each subset. The paper will be updated with the following additional results:
>
> **One-to-Many Generalization Task (avg. performance);** Test on Sora, MorpStudio, Gen2, HotShot, Lavie, Show-, MoonValley, Crafter, ModelScope and WildScrape:
>
> **Training Subset: Pika**
>
>
> | Model              | R        | F1       | AP       |
> |--------------------|----------|----------|----------|
> | NPR                | 0.5144   | 0.5306   | 0.6497   |
> | STIL               | 0.7383   | 0.5165   | 0.6304   |
> | MINTIME-CLIP-B     | 0.2166   | 0.3263   | 0.5738   |
> | FTCN-CLIP-B        | 0.6287   | 0.6423   | 0.7194   |
> | TALL               | 0.7141   | 0.5572   | 0.6225   |
> | XCLIP-B-FT         | 0.6096   | 0.6579   | 0.7831   |
> | DeMamba-XCLIP-FT   | **0.7572** | 0.7263   | **0.8166** |
> | Ours (RestraV)     | 0.7354   | **0.8272** | 0.7967   |
>
>
>
> **Training Subset: SEINE**
>
> | Model              | R        | F1       | AP       |
> |--------------------|----------|----------|----------|
> | NPR                | 0.4618   | 0.5385   | 0.6111   |
> | STIL               | 0.7239   | 0.5057   | 0.6083   |
> | MINTIME-CLIP-B     | 0.6135   | 0.6538   | 0.7443   |
> | FTCN-CLIP-B        | 0.6279   | 0.6991   | 0.8036   |
> | TALL               | 0.6568   | 0.6085   | 0.6805   |
> | XCLIP-B-FT         | 0.7201   | 0.7898   | 0.8880 |
> | DeMamba-XCLIP-FT   | 0.8098   | 0.7873   | **0.8943** |
> | Ours (RestraV)     | **0.8199** | **0.8981** | 0.8543   |
>
> **Training Subset: OpenSora**
>
> | Model              | R        | F1       | AP       |
> |--------------------|----------|----------|----------|
> | NPR                | 0.5929   | 0.5232   | 0.5763   |
> | STIL               | 0.4336   | 0.4892   | 0.5256   |
> | MINTIME-CLIP-B     | 0.6889   | 0.7269   | **0.8286** |
> | FTCN-CLIP-B        | 0.1863   | 0.3005   | 0.5689   |
> | TALL               | 0.4916   | 0.5315   | 0.5710   |
> | XCLIP-B-FT         | 0.6615   | 0.6497   | 0.7154   |
> | DeMamba-XCLIP-FT   | 0.7382   | 0.6713   | 0.7382   |
> | Ours (RestraV)     | **0.7707** | **0.7965** | 0.7167   |
>
> *“Discussion on why the proposed method works is limited”:* We’re happy to expand our current discussion of hypotheses (ll. 295-302, 312-318) in the manuscript within the given page limit. One possible explanation is that SSL models like DINOv2 might develop a strong inductive bias for subtle statistical regularities of the natural world. While video generators like Sora can produce photorealistic videos, they still struggle to perfectly replicate these complex physical dynamics. Even SOTA generators can exhibit temporal inconsistencies or generate implausible physical scenarios. These failures possibly violate the learned statistical regularities of the DINOv2 encoder. In essence, ReStraV works by using an SSL model trained on natural image statistics as an "expert observer" to spot the subtle, unnatural geometric fingerprints left by the imperfect physical simulations of even the most advanced AI video generators.
>
> \[1\] Wang et al. Vidprom: A million-scale real prompt-gallery dataset for text-to-video diffusion models, Neurips, 2024
> \[2\] Ni et al. Genvidbench: A challenging benchmark for detecting ai-generated video, 2025
> \[3\] Motamed et al. Do generative video models understand physical principles?, 2025
> \[4\] Chen et al, DeMamba: AI-Generated Video Detection on Million-Scale GenVideo Benchmark, 2024

---

> > ### Comment · Reviewer_98v6 · 2025-08-03
> > **Additional questions**
> >
> > Thanks to the authors for their detailed responses. It would be great if the authors could comment on the following question.
> >
> > Have the authors considered applying a HVS-inspired system for analyzing perceptual straightness? Please refer to the first question in the original review. The reason for this question is that the original perceptual straightening hypothesis is based on linear-non-linear models of the early HVS regions and their responses to natural videos. This work, however, relies entirely on DL models and therefore does not truly qualify as a "perceptual" approach. Ideally, an "expert observer" is the human visual system. I honestly think that augmenting the current analysis with bandpass filtered systems that mimic LGN/V1 regions of the HVS would strengthen this work. My hunch is that it may even make the system more generalizable.

---

> > > ### Author Response · Authors · 2025-08-04
> > >
> > > Thank you very much for your constructive review and especially for this additional, insightful feedback. We agree that incorporating Human Visual System (HVS)-based models can strengthen the paper. Following your suggestion, we have analyzed the curvature trajectories on the PhysicsIQ videos [1], mirroring the analysis in Figure 2. We tested a diverse new set of models:
> > >
> > > 1) *Gabor Filter Bank* [2]: A classic model mimicking the orientation and spatial frequency selectivity of V1 neurons.
> > > 2) *LGN-V1 Model* [3]: A multi-stage based on [3], which simulates the processing from the Lateral Geniculate Nucleus (LGN) to the primary visual cortex (V1).
> > > 3) Three spatio-temporal models [4-6]: Two video-level embeddings with *3D convolutions* (S3d [4], R3d_18 [5]) and one *multiscale vision transformer* (Mvits_16x4 [6]) following the suggestion of reviewer LGaM.
> > > 4) *Adversarially Trained ResNet-50* [7]: Inspired by work showing the straightening capabilities of robust models [7], we tested an Adversarially Trained ResNet-50 (AdvResNet50).
> > >
> > > | Model | Natural Mean Curvature | GenAI Mean Curvature | Straightening (Pixel - Natural) | AI Detection (GenAI - Natural) |
> > > | :--- | :--- | :--- | :--- | :--- |
> > > | **Pixels** | 99.64 | 93.83 | 0.00 | 5.81 |
> > > | --- | --- | --- | --- | --- |
> > > | DINOv2 | 98.33 | 143.79 | 1.31 | **45.46** |
> > > | CLIP | 99.11 | 136.03 | 0.53 | 36.92 |
> > > | BYOL | 97.49 | 132.81 | 2.15 | 35.32 |
> > > | SimCLR | 101.21 | 128.56 | -1.57 | 27.35 |
> > > | SINResNet50 | 97.70 | 119.95 | 1.94 | 22.25 |
> > > | ResNet50 | 103.96 | 119.82 | -4.32 | 15.86 |
> > > | VGG | 99.99 | 115.81 | -0.35 | 15.82 |
> > > | AlexNet | 105.02 | 117.90 | -5.38 | 12.88 |
> > > | **New Models** | | | | |
> > > | MViT_16x4 | 101.03 | 101.97 | -1.39 | 0.94 |
> > > | S3d | 95.81 | 96.25 | 3.83 | 0.44 |
> > > | R3d_18 | 95.12 | 95.23 | 4.52 | 0.11 |
> > > | AdvResNet50 | 90.48 | 86.78 | 9.16 | -3.70 |
> > > | GFB | **83.95** | 70.90 | **15.69** | -13.06 |
> > > | LGN-V1 | 90.58 | **67.29** | 9.06 | -23.29 |
> > >
> > > **TL;DR:** These results reveal a crucial insight: while the HVS-inspired models (GFB, LGN-V1) are most effective at the classic *"Perceptual Straightening"* task, they do so for both natural and AI-generated videos, which paradoxically makes them *less* effective for AI Detection than top SSL models like DINOv2.
> > >
> > > First, we wish to emphasize the critical distinction between *"Perceptual Straightening"* and *"AI Detection."*  This is one of the key concepts of our paper that differs from prior work on straightening. A model can be ineffective at the classic straightening task (i.e., its representation of natural videos is more curved than in the pixel domain), yet still be an excellent AI detector if the *magnitude* of the delta, |Δ (GenAI Curvature - Natural Curvature)|, is large. Our analyses confirm that the correlation between Straightening and AI Detection (|Δ|) is *⍴=-0.1313* with a non-significant *p=0.6408*. This demonstrates that while both tasks are derived from the same curvature metric, they can become distinct.
> > >
> > > Thinking further, this suggests that there is a new phenomenon with modern, powerful vision encoders: They do not improve on the classical straightening task by Hénaff et al [2]. But they do, selectively, learn representations with higher curvature only for GenAI videos, likely because the temporal artifacts of current generators violate their strong inductive biases for natural world statistics. As the table shows, the HVS-inspired models (and adversarially trained Resnet50) are the *only models that produce a negative delta*, with the LGN-V1 model being the most effective “straightener” of both natural and AI generated videos. The spatio-temporal models are smoothing out the difference between Natural and AIGen.
> > >
> > > Thank you again for the excellent suggestions that led to this analysis. We will include the analysis of these findings in a revised Figure 2 and add the LGN-V1 model to our main experiments in Sections 4 and 5, thereby ensuring we evaluate the best discriminators from both the classic HVS and modern DL domains.
> > > Given that your initial score is the only one leaning towards rejection, could you kindly let us know whether we’ve been able to address your suggestions and concerns?
> > >
> > > [1] Saman M. et al. Do generative 408 video models understand physical principles?, 2025.
> > >
> > > [2] Jones JP, Palmer LA. An evaluation of the two-dimensional Gabor filter model of simple receptive fields in cat striate cortex. J Neurophysiol. 1987
> > >
> > > [3] Olivier J Hénaff et al. Perceptual straightening of natural videos. Nature neuroscience, 2019.
> > >
> > > [4] Xie, S. et al. (2018). Rethinking Spatiotemporal Feature Learning: Speed-Accuracy Trade-offs in Video Classification.
> > >
> > > [5] Hara, K. et al. (2018). Can Spatiotemporal 3D CNNs Retrace the History of 2D CNNs and ImageNet?. IEEE CVPR.
> > >
> > > [6] Fan, H. et al. (2021). Multiscale Vision Transformers. In Proceedings of the IEEE/CVF.
> > >
> > > [7] Harrington A. et al. Exploring perceptual straightness in learned visual representations. ICLR, 2023.

---

> > > > ### Comment · Reviewer_98v6 · 2025-08-05
> > > > **Concluding Remarks**
> > > >
> > > > The analysis is appreciated. These insights are interesting and deserve further investigation in future work. Overall, the responses and the clarifications have certainly helped, and I hope that these are reflected in the final version of the paper. Based on the discussion, I will raise my score.

---

> > > > > ### Author Response · Authors · 2025-08-07
> > > > >
> > > > > Thank you again for your comments, they have greatly improved the clarity of the manuscript!

---

### Official Review · Reviewer_LGaM · 2025-07-03

**Clarity:** 3
**Significance:** 2
**Originality:** 4
**Rating:** 4
**Confidence:** 4

**Summary:**

This paper introduces ReStraV, a novel method to distinguish natural videos from AI-generated ones. It utilizes the "perceptual straightening" hypothesis, observing that real video trajectories are straighter in latent space than synthetic ones. ReStraV computes temporal curvature and stepwise distance between frame embeddings of a pre-trained image encoder (DINOv2), which are used to train a lightweight classifier. The approach is computationally efficient, offering a robust and low-cost solution to detect synthetic content in videos.

**Questions:**

- I’m curious to know how a video-level embedding with 3D convs or transformers interacts with the straightening observation. For example, whether the artifacts in synthetic videos are smoothed out? If so, would the straightening classifier still work?
- Why include the CLS token? It seems to have a different nature than patch embeddings.
- In section 6, how was the “seven distance values” chosen? Why 7?
- It is known that a vanilla deep vision encoder does not straighten (meaning its latent space is not straighter than the pixel space), but models that are adversarially trained, or directly trained on the straightening loss, can produce much straighter trajectories. Would these models work better or worse for the classifier?

**Ethical Concerns:**

["NO or VERY MINOR ethics concerns only"]

**Final Justification:**

My score will be maintained based on my initial reviews and the quality of the author's rebuttal.

**Limitations:**

This is a very simple approach, so the significance comes from the practical usage. However, I worry that pairwise distance and curvature is just a temporary “bias” of those video generative models rather than an intrinsic property. For example, they are often prompted to generate dynamic scenes (few prompts will ask the model to output a near stationary video), or they often fail at object persistence, which may have given rise to the larger frame-wise distance. If the prompt is carefully designed to generate slow or little motion, would the classifier still work?

**Quality:**

3

**Strengths And Weaknesses:**

Strengths:
- This is a very simple, light-weighted, effective method for detecting AI generated videos. Simplicity here is a pro and not meant to be a criticism.

Weaknesses:
- The paper can benefit greatly from a robustness analysis. I’m interested to know whether the classifier is robust to 1) choice of vision encoder 2) the time interval between frames on which distance and curvature are computed 3) duration of the video 4) types of artifacts in synthetic videos 5) the absence/presence of any one of the 21 features that are put into the classifier (which feature plays the most important role?)
- Natural videos often have shot transitions. I would assume the straightening hypothesis only holds within one shot, or a short period of time. This constraints the scope of the method.

Suggestions for improvement:
- It would be nice to show a few examples of real videos and their “synthetic replicas”, to let readers have an idea of 1) the common types of artifacts in synthetic videos 2) how realistic the synthetic videos look like, so that we can have a sense of how difficult the detection problem is.

---

> ### Author Rebuttal · Authors · 2025-07-31
>
> First of all, thank you very much for highlighting and appreciating the simplicity and effectiveness of our method.
>
> *“The paper can benefit greatly from a robustness analysis.”:* We agree that a robustness analysis would greatly benefit the paper.
>
> *(1) Choice of vision encoder.* We would like to draw attention to Figure 2b where we systematically tested eight encoders (CNNs, Transformers, Contrastive) and selected DINOv2 due to its ability to separate natural and AI video curvature.
>
> *(2\) Time interval between frames* and *(3) duration of the video.* Please refer to the "Supplementary Material: zip" folder on the submission page where you will find the impact of *video length* and *sampling density/interval between frames* on performance and inference time (Figure 1\) and the impact of *temporal window position* (Figure 2).
>
> *(4)* ReStraV’s is intentionally artifact-agnostic. Instead of targeting specific artifacts (e.g., GAN-specific frequency patterns or face-swapping irregularities) it quantifies a more fundamental deviation from natural world dynamics. We hypothesize that models struggle to replicate natural video's smooth temporal structure, leading to higher curvature. This is validated by ReStraV's strong performance across diverse AI generators (Tables 2-4) and its success on the Physics-IQ test.
>
> *(5) Feature Ablation:* Following your suggestion we performed a permutation importance analysis, training our MLP classifier on 50,000 AI-generated samples from VideoProM and 50,000 natural videos from DVSC2023 (Section 6):
>
> | Feature         | Mean F1-score drop (%) | Standard Deviation (%) |
> |-----------------|------------------------|------------------------|
> | curvature\_mean | 8.56%                  | 0.31%                  |
> | curvature\_var  | 3.11%                  | 0.17%                  |
> | distance\_min   | 1.53%                  | 0.09%                  |
> | distance\_max   | 1.31%                  | 0.11%                  |
> | curvature\_max  | 0.65%                  | 0.05%                  |
> | distance\_0     | 0.61%                  | 0.07%                  |
> | distance\_min   | 0.51%                  | 0.07%                  |
> | distance\_mean  | 0.09%                  | 0.04%                  |
> | distance\_var   | 0.11%                  | 0.13%                  |
> | curvature\_min  | 0.06%                  | 0.04%                  |
> | distance\_5     | 0.09%                  | 0.02%                  |
> | curvature\_1    | 0.08%                  | 0.03%                  |
> | distance\_2     | 0.07%                  | 0.04%                  |
> | curvature\_2    | 0.07%                  | 0.02%                  |
> | distance\_4     | 0.06%                  | 0.01%                  |
> | curvature\_0    | 0.05%                  | 0.05%                  |
> | curvature\_5    | 0.04%                  | 0.03%                  |
> | curvature\_6    | 0.03%                  | 0.05%                  |
> | curvature\_3    | 0.03%                  | 0.03%                  |
> | curvature\_4    | 0.02%                  | 0.01%                  |
> | distance\_1     | 0.02%                  | 0.02%                  |
> | distance\_3     | 0.15%                  | 0.03%                  |
> | distance\_6     | 0.19%                  | 0.04%                  |
>
> Mean curvature is the most critical feature, followed by curvature variance. This validates Figure 5, where features with the clearest separation between natural and AI video distributions proved most important.
>
> *“Natural videos often have shot transitions.”:* Your point about shot transitions is well-taken. There is, indeed, no reason to expect the straightening hypothesis to hold across shots. Presumably, shot transitions will come with a high curvature. This affects both shot transitions in natural and AI generated videos. If the latter have, on average, more shot transitions, then this might explain part of the performance of our classifier. Using "scenedetect"  on 13,000 AI-generated and 13,000 natural videos, AI videos averaged 5.89 seconds shot duration, while natural videos averaged 7.04 seconds:
> | Model        | Average Duration (seconds) | Average # of scene cuts in first 2s |
> |--------------|----------------------------|-------------------------------------|
> | PIKA         | 2.97s                      | 0.00                                |
> | Show1        | 3.62s                      | 0.00                                |
> | ModelScope   | 3.84s                      | 0.00                                |
> | Gen2         | 4.00s                      | 0.00                                |
> | HotShot      | 4.60s                      | 0.01                                |
> | Crafter      | 5.43s                      | 0.00                                |
> | MoonValley   | 5.59s                      | 0.00                                |
> | T2vz         | 6.01s                      | 0.00                                |
> | Vc2          | 6.98s                      | 0.10                                |
> | **Natural** | **7.04s** | **0.08** |
> | MorphStudio  | 7.05s                      | 0.02                                |
> | OpenSora     | 8.22s                      | 0.04                                |
> | Sora         | 12.40s                     | 0.04                                |
>
> We use a relatively short, 2s window, which reduces the probability of encountering a hard scene cut. Our ablation study (Supplementary Material, Figure 2) confirms detection performance is robust regardless of window position, indicating insensitivity to local video dynamics like shot transitions. This is further supported by promising results on models generating longer scenes (Sora, OpenSora, MorphStudio). We will add this important discussion to the paper.
>
> *“It would be nice to show a few examples of real videos and their synthetic replicas”:* Thanks for suggesting example frames of real and synthetic videos for the appendix. In Section A.10.3 “Qualitative Examples of Plants Task” we directly compare frames from natural videos (HD-VG/130M) with their synthetic counterparts from generators like Mora, MuseV, SVD, and CogVideo. In addition, we will add randomly sampled frames across tasks to add more diverse examples.
>
> **Miscellaneous Questions:**
>
> * *“Models that are adversarially trained, or directly trained on the straightening loss, can produce much straighter trajectories. Would these models work better or worse for the classifier?”:* Thank you for the insightful points about adversarial models. *S*traightness is task-dependent and not universally desirable. For example, a video prediction model must match the input video's curvature rather than minimize it as showed in Harrington et al.'s 'Exploring Perceptual Straightness in Learned Visual Representations' (ICLR 2023). We will add this discussion to the paper to highlight that our goal is not pure straightening but relative straightening (of natural vs. synthetic videos) which enables discriminating the two.
>
> * *“Why include the CLS token?”:* The CLS token summarizes the entire frame, while patch embeddings capture local details, enabling the model to identify both large-scale and fine-grained inconsistencies. We will clarify this in the text.
> * “In section 6, how was the “seven distance values” chosen? Why 7?”: We considered 24 frames with a sampling interval of 3 (8 final frames in total) from a fixed 2-second window, *selected as the shared duration across all datasets and generators to avoid bias and missing data* (see Section 4\). This choice also optimizes detection accuracy and computational efficiency, as shown in our ablation studies (Supplementary Material). We will further justify this in the paper.
>
> *“I worry that pairwise distance and curvature is just a temporary “bias” of those video generative models rather than an intrinsic property.”*  Great questions. We have three pieces of evidence that suggest the geometric signal used by ReStraV is not just a temporary bias but, somewhat, robust to future models:
>
> * *Performance on a New SOTA Model Veo3 (Out-of-distribution):* In response to your concern, we tested our method on videos from the new Veo3 model.  This model is considered to not typically suffer from poor object persistence. A simple Linear Regression trained with the model and futures as in Section 5 when tested on 100 Veo3 and 100 natural videos achieve: ACC \= 0.832 | F1 \= 0.851 | AUROC \= 0.869. We expect performance to improve with a larger training set including Veo3 videos.
> * *Generalization to Future Models:* We tested this exact scenario. As shown in Table 3 (and in the previous point with VEO3), ReStraV was trained on older generators and then tested on videos from Sora, a more advanced "future generator". The method achieved 92.85% AUROC. While not perfect, those results suggest a degree of future-proofing.
> * *Effectiveness on Low-Motion Content:* We tested our method on the GenVidBench "Plants" task, its most challenging subset of GenVidBench. And we argue that it is since it often features subtle movements like wind and complex, stochastic patterns rather than dynamic action. As shown in Table 4(b), ReStraV achieved an ACC. of over 93%. This could indicate ReStraV detects unnatural patterns even in the presence of slow or little motion. While no detection method is guaranteed to be permanent, these results suggest that the geometric trajectory analysis of ReStraV captures a more fundamental property of AI-generated videos.

---

> > ### Comment · Reviewer_LGaM · 2025-08-05
> >
> > I appreciate the responses given by the authors. I agree that this is a reliable and useful method given the robustness analysis. That being said, I expect the paper to reveal more intuition in terms of "why this works" and not just "we tried all these models and this one works". Why does DINOv2 work better than other image embeddings? What's the intuition behind the distinction between natural VS AI videos that make them straighter in the DINOv2 space? This is the main reason why I remain my current score.
> > I still think this is a useful approach and should be accepted.

---

> ### Author Response · Authors · 2025-08-06
> **Intuition on "why this works"**
>
> Thanks for getting back to us, we're glad to hear the rebuttal improved your assessment and you're in favor of acceptance. We have some hypotheses regarding an intuition for why SSL works better than other models. Should you have a specific experiment in mind, please let us know.
>
> Our guiding hypothesis is that SSL models like DINOv2 might develop a strong inductive bias for subtle statistical regularities of the natural world. While video generators like Sora can produce photorealistic videos, they still struggle to perfectly replicate these complex physical dynamics. Even SOTA generators can exhibit temporal inconsistencies or generate implausible physical scenarios [1]. These failures possibly violate the learned statistical regularities of the DINOv2 encoder.
>
> To test our hypothesis that DINOv2 detects violations of natural regularities rather than simply enforcing generic straightening, we analyzed curvature trajectories on the PhysicsIQ dataset [1], comparing classic SSL models to a diverse set of new architectures. We included Human Visual System (HVS)-inspired models (Gabor Filter Bank [2], LGN-V1 [3]), spatio-temporal video models (S3d [4], R3d_18 [5], MViT [6]), and an adversarially trained ResNet-50 [7]. The results are summarized below:
>
> | Model | Natural (Mean Curvature) | GenAI (Mean Curvature) | Straightening (Pixel-Nat) | AI Detection (GenAI-Nat) |
> |:---|:---:|:---:|:---:|:---:|
> | Pixel | 99.64 | 93.83 | 0.00 | 5.81 |
> | DINOv2 | 98.33 | 143.79 | 1.31 | **45.46** |
> | CLIP | 99.11 | 136.03 | 0.53 | 36.92 |
> | BYOL | 97.49 | 132.81 | 2.15 | 35.32 |
> | SimCLR | 101.21 | 128.56 | -1.57 | 27.35 |
> | SIN-ResNet50 | 97.70 | 119.95 | 1.94 | 22.25 |
> | ResNet50 | 103.96 | 119.82 | -4.32 | 15.86 |
> | VGG | 99.99 | 115.81 | -0.35 | 15.82 |
> | AlexNet | 105.02 | 117.90 | -5.38 | 12.88 |
> **New Models**
> | Mvits_16x4 | 101.03 | 101.97 | -1.39 | 0.94 |
> | S3d | 95.81 | 96.25 | 3.83 | 0.44 |
> | R3d_18 | 95.12 | 95.23 | 4.52 | 0.11 |
> | AdvResNet50 | 90.48 | 86.78 | 9.16 | -3.70 |
> | GFB | **83.95** | 70.90 | **15.69** | -13.06 |
> | LGN-V1 | 90.58 | **67.29** | 9.06 | -23.29 |
>
> As the table shows, there is an interesting inverse relationship: the HVS-inspired models (LGN-V1, GFB) excel at the classic straightening task (highest Straightening values), but they apply this smoothing to both natural and AI-generated videos, making them poor detectors (as shown by their negative AI Detection delta).
> Conversely, DINOv2 is a superior AI detector precisely because it is not a simple straightener. Its representation is uniquely sensitive to the unnatural artifacts in AI videos, which cause a massive increase in trajectory curvature (the largest positive AI Detection delta).
>
> In essence, DINOv2 acts as a high-level "expert observer" for natural world physics, while the HVS models are low-level processors that merely enforce smoothness. Spatio-temporal models like MViT appear to smooth the temporal signal too aggressively, erasing the very artifacts that DINOv2 leverages for detection.
>
> We will integrate this entire discussion, the expanded table, and a revised Figure 2  in the final manuscript (ll. 295-302, 312-318).
>
> [1] Saman M. et al. Do generative 408 video models understand physical principles?, 2025.
>
> [2] Jones JP, Palmer LA. An evaluation of the two-dimensional Gabor filter model of simple receptive fields in cat striate cortex. J Neurophysiol. 1987
>
> [3] Olivier J Hénaff et al. Perceptual straightening of natural videos. Nature neuroscience, 2019.
>
> [4] Xie, S. et al. (2018). Rethinking Spatiotemporal Feature Learning: Speed-Accuracy Trade-offs in Video Classification.
>
> [5] Hara, K. et al. (2018). Can Spatiotemporal 3D CNNs Retrace the History of 2D CNNs and ImageNet?. IEEE CVPR.
>
> [6] Fan, H. et al. (2021). Multiscale Vision Transformers. In Proceedings of the IEEE/CVF.
>
> [7] Harrington A. et al. Exploring perceptual straightness in learned visual representations. ICLR, 2023.

---

### Author Response · Authors · 2025-08-07
**Author's summary of rebuttal discussion**

We would like to express our sincere gratitude to all the reviewers for a detailed and constructive review process.
We were very encouraged by their positive assessment of our work, describing it as an *“innovative approach”* with *“simplicity and effectiveness”* (**NU9X, and GaM**), a method that is *“lightweight and delivers good performance”* (**98v6**), and a paper that is *“clearly written”* with *“strong empirical results”* (**R-YPpG**).

The discussion period allowed us to address the following points, among others:

A key theme, brought up by **R-LGaM** and **R-98v6**, was the request for a *deeper intuition for why our method is effective*. To investigate this, we conducted a new analysis comparing a diverse set of architectures (SSL models, HVS-inspired systems, spatio-temporal models, and adversarially trained networks). The results offered a compelling explanation: *our method's strength lies not in simple straightening*, but in DINOv2's *strong inductive bias for the regularities of the natural world.* The crucial insight is that trajectory curvature, inspired by the perceptual straightening hypothesis, provides a remarkably *simple metric to quantify when AI-generated videos violate this learned bias.* These violations manifest as a detectable increase in curvature, making DINOv2's representation space uniquely effective for this task.

We also worked to strengthen the paper's empirical validation based on further feedback. We included new benchmark results against additional state-of-the-art baselines and demonstrated robustness with a *zero-shot test on the new Veo3 model* (**R-LGaM, and R-YPpG**). As requested from **R-LGaM and R-NU9X**,  we also performed a full feature importance analysis and discussed the method's inherent robustness to noise. These efforts expanded our evaluation to a comprehensive total of 24 unique baselines and 18 distinct AI generators.

Finally, to address the *"possible circumvention"* concern from **R-YPpG**, we provided new data showing a moderate negative correlation between our curvature metric and video fidelity. This suggests that *circumventing our detector might not be a non-trivial challenge*, though we certainly acknowledge the inevitable *"arms race"* in this domain.


*We will ensure that all new results and discussions are carefully integrated into the final manuscript.*

---

### Decision · Program_Chairs · 2025-09-17

**Decision:**

Accept (poster)

**Comment:**

The paper proposes ReStrav for detecting AI-generated videos. The contribution exploits a perceptual straightening hypothesis - that real videos tend to follow straighter trajectories in a DINOv2 representation space than synthetic ones.  Some interesting results are presented including on unseens models (Veo3) indicating this may be a generally useful detection feature non-specific to learned data-specific priors.

The paper has received unanimous recommendation to accept in the final, post-rebuttal discussion stage (2x firm accept, 2x borderline accept).

NU9X is generally positive and considers the insight valuable.

98v6 was initially borderline and had doubts re: embedding model choice but was convinced by the rebuttal experiments and raised their rating.

LGam and YPpH are both positive but wanted to see more theory vs. empirical observation of the phenomenon.  This was not really provided in the rebuttal but some diversity of model choice was explored and greater explanation / conclusion offered.

Overall the AC considers this an interesting and insightful paper on AI video detection, and agrees with the unanimous decision of the reviewers to accept the paper.